# Methane emissions from China: a high-resolution inversion of TROPOMI satellite observations

Zichong Chen[1*], Daniel J. Jacob[1], Hannah Nesser[1], Melissa P. Sulprizio[1], Alba Lorente[2], Daniel J. Varon[1], Xiao Lu[3], Lu Shen[4], Zhen Qu[1], Elise Penn[1], and Xueying Yu[5]

[1]Department of Earth and Planetary Sciences, Harvard University, Cambridge, MA, USA

[2]SRON Netherlands Institute for Space Research, Leiden, the Netherlands

[3]School of Atmospheric Sciences, Sun Yat-sen University, Zhuhai, Guangdong, China

[4]Department of Atmospheric and Oceanic Sciences, School of Physics, Peking University, Beijing, China

[5]Department of Soil, Water, and Climate, University of Minnesota, Saint Paul, MN, USA

10 *Correspondence to*: Zichong Chen (zchen1@g.harvard.edu)

**Abstract.** We quantify methane emissions in China and the contributions from different sectors by inverse analysis of 2019 TROPOMI satellite observations of atmospheric methane. The inversion uses as a prior estimate the latest 2014 national sector-resolved anthropogenic emission inventory reported by the Chinese government to the United Nations Framework Convention on Climate Change (UNFCCC) and thus serves as a direct evaluation of that inventory. Emissions are optimized with a Gaussian mixture model (GMM) at up to $0.25^{\circ} \times 0.3125^{\circ}$ resolution. The optimization is done analytically assuming lognormally distributed errors on prior emissions. Errors and information content on the optimized estimates are obtained directly from the
20 analytical solution and also through a 36-member inversion ensemble. Our best estimate for total anthropogenic emissions in China is 65.0 (57.7-68.4) Tg a$^{-1}$, where parentheses indicate the uncertainty range determined by the inversion ensemble. Contributions from individual sectors include 16.6 (15.6-17.6) Tg a$^{-1}$ for coal, 2.3 (1.8-2.5) for oil, 0.29 (0.23-0.32) for gas, 17.8 (15.1-21.0) for livestock, 9.3 (8.2-9.9) for waste, 11.9 (10.7-12.7) for rice paddies, and 6.7 (5.8-7.1) for other sources. Our estimate is 21% higher than the Chinese inventory reported to the UNFCCC (53.6 Tg a$^{-1}$), reflecting upward corrections to emissions from oil (+147%), gas (+61%), livestock (+37%), waste (+41%), and rice paddies (+34%), but downward correction for coal (-15%). It is also higher than previous inverse studies (43-62 Tg a$^{-1}$) that used the much sparser GOSAT satellite observations and were conducted at coarser resolution. We are in particular
better able to separate coal and rice emissions. Our higher livestock emissions are attributed largely to northern China where GOSAT has little sensitivity. Our higher waste emissions reflect at least in part a rapid growth in wastewater treatment in China. Underestimate of oil emissions in the UNFCCC report appears to reflect unaccounted-for super-emitting facilities. Gas emissions in China are mostly from distribution, in part because of low emission factors from production and in part because 42% of the gas is imported. Our estimate of emissions per unit of domestic gas production indicates a low life-cycle loss rate of 1.7 (1.3-1.9) %, which would imply net climate benefits from the current coal-to-gas energy transition in China. However, this small loss rate is somewhat misleading considering China's high gas imports, including from Turkmenistan where emission per unit of gas production is very high.

## 1. Introduction

Methane ($CH_4$) is a potent greenhouse gas with an atmospheric lifetime of $9.1 \pm 0.9$ years (*Prather et al*., 2012). Its atmospheric concentration has nearly tripled since pre-industrial times because of anthropogenic emissions. The resulting radiative forcing from methane on an emission basis (including effects on tropospheric ozone, stratospheric water vapor, and carbon dioxide ($CO_2$)) is $1.21$ W m$^{-2}$ since the pre-industrial era, compared to $2.16$ W m$^{-2}$ for $CO_2$ *(Naik et al*, 2021). Reducing methane emissions is a recognized priority under the Paris Agreement. As of January 2022, 111 countries have signed the Global Methane Pledge to reduce their methane emissions by 30% below 2020 levels by 2030 (*https://www.globalmethanepledge.org*).

China is the single largest contributor to global anthropogenic methane emissions (*Worden et al*., 2022). It is estimated to have emitted 46-74 Tg a$^{-1}$ out of a global anthropogenic source of 349-393 Tg a$^{-1}$ for the 2008-2017 decade (*Saunois et al*., 2020). According to the latest national emission inventory for 2014 submitted by the Chinese government to the United Nations Framework Convention on Climate Change (*UNFCCC*, 2020), China emitted 53.6 Tg a$^{-1}$ including contributions from coal mining (38%), livestock (24%), rice paddies (16%), landfills (7%), wastewater management (5%), oil/gas systems (2%), and other sources (8%). Emission inventories reported to the UNFCCC are 'bottom-up' estimates derived from activity data and emission factors (EFs) per unit of activity, supplemented in some cases with more source-specific information. There are large uncertainties in these bottom-up estimates (*Saunois et al*.,
2020; *Gao et al*., 2021). Different bottom-up national inventories for China as reported by *Lin et al*. (2021) ranged from 44.4 to 57.5 Tg a$^{-1}$ in 2010, with larger relative differences for individual sectors. These uncertainties make it difficult to set targets for reducing methane emissions.

The recent 'coal-to-gas' transition policy in China (*Qin et al*., 2018) has raised growing awareness of oil/gas methane emissions, which are presently small but could grow rapidly. Gas is projected to account for 15% of total energy supply in China by 2030 (*Gan et al*., 2020). It is crucial to quantify China's oil/gas and coal methane emissions in order to assess the climate benefits of switching from coal to gas (*Alvarez et al*., 2012, 2018). Chinese oil/gas emissions in the most recent version of the widely used bottom-up EDGAR v6 inventory for 2018 (3.4 Tg a$^{-1}$; *Crippa et al*., 2021) are much higher than in the government report to the UNFCCC (1.1 Tg a$^{-1}$),
while coal emissions in EDGAR v6 (20.4 Tg a$^{-1}$) are consistent with the government report (19.5 Tg a$^{-1}$). Previous versions of EDGAR overestimated coal emissions from China (*Bergamaschi et al*., 2013; *Turner et al*., 2015).

Satellite observations of atmospheric methane in the shortwave infrared (SWIR) offer important 'top-down' information for evaluating bottom-up inventories and reducing uncertainty (*Jacob et al*., 2016). Exploiting this information involves inversion of the observations with an atmospheric transport model relating emissions to atmospheric concentrations, and using the bottom-up inventory as prior information (*Brasseur and Jacob*, 2017). A number of global and regional inversions relevant to China have been conducted with satellite observations from the Scanning Imaging Absorption Spectrometer for Atmospheric Chartography (SCIAMACHY) for
2003–2012 (*Bergamaschi et al*, 2013; *Houweling et al*., 2014) and the Greenhouse Gases Observing Satellite (GOSAT) for 2009-present (*Alexe et al*., 2015; *Turner et al., 2015*; *Pandey et al*., 2016; *Miller et al*., 2019; *Maasakkers et al*., 2019; *Y. Zhang et al*., 2021; *Qu et al., 2021*; *Deng et al*., 2022). The TROPOspheric Monitoring Instrument (TROPOMI) satellite instrument launched in October 2017 provides global daily data with 5.5 km × 7 km (7 km × 7 km before August 2019) pixel resolution, considerably increasing coverage relative to previous satellite instruments (*Hu et al*., 2018; *Lorente et al*., 2021). Recent studies have used TROPOMI data in

inverse analyses of methane emissions for North America (*Y. Zhang et al.*, 2020; *Shen et al.*, 2021) and globally at coarse resolution (*Qu et al.*, 2021; *van Peet et al.*, 2021). *Qu et al.* (2021) pointed out that their TROPOMI inversion suffered from major artifacts in southern China due to mislocation of prior coal emissions, juxtaposition of coal and rice emissions at the ~200 km resolution of the inversion, and extensive seasonal cloudiness.

Here we use TROPOMI observations for 2019 to quantify methane emissions from China at up to 0.25º ×0.3125º (~ 25 × 25 km$^2$) resolution and with attribution to different source sectors. Our inversion uses the Chinese national inventory reported to the UNFCCC as prior information so that our results are directly relevant for evaluating that inventory, and includes an improved prior spatial distribution of methane emissions from the coal sector (*Sheng et al.*, 2019). We apply an analytical solution to the Bayesian inference of methane emissions (*Jacob et al.*, 2016), which has the advantage of providing closed-form error statistics and hence information content as part of the solution, and also allows us to conduct an ensemble of sensitivity inversions at minimal added computational effort.

## 2. Data and Methods

We conduct the inversion of TROPOMI data for the full year of 2019 over the East Asia domain of Fig. 1 (15º-55º N, 70º-140º E) at up to 0.25º×0.3125º resolution. In this section we describe the TROPOMI observations (Sect. 2.1), the prior emission estimates (Sect. 2.2), the GEOS-Chem chemical transport model used as forward model for the inversion (Sect. 2.3), the analytical inversion method (Sect. 2.4), the sectoral attribution of inversion results (Sect. 2.5), and the ensemble of sensitivity inversions (Sect. 2.6).

2.1 TROPOMI observations

TROPOMI is onboard the polar sun-synchronous Sentinel-5 Precursor satellite with a ~ 13:30 local overpass time. The instrument observes methane columns by solar backscatter in the 2.3 μm absorption band with near-uniform sensitivity down to the surface. The column-averaged dry-air methane mixing ratio (XCH$_4$) is retrieved with a full-physics algorithm (*Butz et al.*, 2011) together with surface and atmospheric scattering properties. We use the recently updated TROPOMI version 2.02 retrieval from the Netherlands Institute for Space Research (*Lorente et al.*, 2021; *http://www.tropomi.eu/data-products/methane*), filtering out low-quality retrievals ('qa_value' < 0.5) and surfaces above 2 km where the stratospheric contribution to the column is large (*Shen et al.*, 2022). We further adopt the 'blended albedo' filter suggested in *Lorente et al* (2021) to remove snow- or ice-covered scenes identified by blended albedo exceeding 0.8 from October to April.

Global mean bias in the TROPOMI observations is inconsequential for regional inversions because it can be incorporated in the boundary conditions (defined as the edges of the study domain), and random error (precision) is effectively reduced through the large number of observations (Fig. 1). More problematic is spatially variable bias, which corrupts the information on methane concentration gradients used to optimize emissions in the inversion. This variable bias typically arises from aliasing of surface spectral features into the XCH$_4$ retrieval. *Lorente et al.* (2021) estimated a variable bias of 5.6 ppb for the TROPOMI XCH$_4$ full-physics retrieval as the spatial standard deviation of the mean difference with ground-based methane observations from the Total Carbon Column Observing Network (TCCON; *Wunch et al.*, 2011). This is below

the threshold requirement of 10 ppb recommended by *Buchwitz et al*. (2015) for use of satellite data in regional inversions. However, the TCCON network is sparse and includes no sites in China.

*Qu et al*. (2021) conducted a more thorough worldwide analysis of variable bias in the TROPOMI version 1.03 data (*Hu et al*., 2018) by using the GOSAT observations as reference on a $4^o \times 5^o$ grid. GOSAT is much less subject to retrieval artifacts because of its higher spectral resolution and its use of the $CO_2$ proxy retrieval method in the 1.65 μm absorption band (*Parker et al*., 2020). *Qu et al*. (2021) found TROPOMI variable biases typically in the range 9-13 ppb but exceeding 20 ppb for some regions. Repeating this analysis for our East Asia domain with

TROPOMI version 2.02 (*Lorente et al*., 2021) on the $0.25^o \times 0.3125^o$ grid, we find a mean TROPOMI-GOSAT difference of -9.9 ± 17.6ppb (Fig. 2a). The mean difference is largely driven by TROPOMI values below 1830 ppb at high latitudes (Fig. 1), likely reflecting snow-covered surfaces that are not successfully removed by the blended albedo filter. The standard deviation of the difference (measure of variable bias) is relatively high in part due to the high spatial resolution in our analysis, which also means that a higher bias threshold is acceptable because methane enhancements are larger. Here we exclude TROPOMI observations that show discrepancies larger than 20 ppb compared to GOSAT. The mean TROPOMI-GOSAT difference after these outlying data have been excluded is -3.6 ± 9.1 ppb with no evident regional structure (Fig. 2b).

Fig. 1 shows the mean TROPOMI observations for 2019 retained in our analysis on the $0.25^o \times 0.3125^o$ grid, along with the number of observations in each grid cell. We assimilate $m = 5907939$ TROPOMI retrievals over the inversion domain. There are few observations in western China and no observations at all in Tibet because we have excluded locations with surface altitude above 2 km following *Shen et al*. (2022).

2.2 Prior emissions

Fig. 3 shows the prior estimates of emissions from different sectors over the inversion domain and Table 1 gives national totals for China. Anthropogenic emissions for China are from the

latest 2014 national governmental report to the UNFCCC (*UNFCCC*, 2020). Emissions from coal and oil/gas exploitation are spatially allocated to the $0.25^o \times 0.3125^o$ GEOS-Chem grid using infrastructure information compiled by the Global Fuel Exploitation Inventory (GFEI v2; *Scarpelli et al*., 2022). This includes bottom-up information from *Sheng et al* (2019) for the distribution of China's coal emissions. Other anthropogenic sources are spatially allocated using the EDGAR v4.3.2 inventory. Anthropogenic emissions outside of China are from GFEI v2 for fuel exploitation and from EDGAR v4.3.2 for other sectors. *Wolf et al*. (2017) produced an alternative global gridded inventory for livestock emissions but we find that it is too uniform over China, as *Scarpelli et al*. (2020a) previously found over Mexico, because they use livestock numbers resolved only by province and distribute them over all grasslands and shrublands. All

anthropogenic emissions are assumed to be aseasonal, except for manure management for which we apply temperature-dependent corrections following *Maasakkers et al*., (2016) and rice paddies for which we apply seasonal corrections derived from a biogeochemical model (*B. Zhang et al*., 2016).

Wetland emissions are monthly means for 2019 on a 0.5º×0.5º grid from the nine-member high-performance subset of the WetCHARTs v1.3.1 inventory ensemble that best fits global GOSAT inversions (*Ma et al*, 2021). Other natural sources include daily open-fire emissions from the Global Fire Emissions Database version 4s (GFED4s; *van der Werf et al*., 2017), termite emissions from *Fung et al* (1991), and geological seepage emissions from *Etiope et al* (2019) scaled to a global magnitude of 2 Tg $a^{-1}$ following *Hmiel et al* (2020).

### 2.3 GEOS-Chem chemical transport model

A nested version of the GEOS-Chem chemical transport model (13.0.0; *https://doi.org/10.5281/zenodo.4618180*) is used as the forward model in the inversion to relate methane emissions to atmospheric observations. The model is driven by GEOS-FP reanalysis meteorological fields with 0.25º ×0.3125º spatial resolution and 3-hour temporal resolution (1-h for mixing depths and surface fields) from the NASA Global Modeling and Assimilation Office (*Lucchesi*, 2013). We conduct GEOS-Chem model simulations at 0.25º ×0.3125º resolution over the study domain of Fig. 1 for 2019. The nested version of GEOS-Chem is similar to that used in previous regional inversions of TROPOMI observations (Y. *Zhang et al*., 2020; *Shen et al*., 2021) and uses 3-hour dynamic boundary conditions from the global GEOS-Chem simulated vertical profiles at 2º× 2.5º resolution for 2019 with posterior methane emissions optimized by TROPOMI observations (*Qu et al*., 2021). The global simulation includes methane sinks from atmospheric oxidation and uptake by soils, but these are inconsequential in the nested version because the ventilation time scale for the nested domain is much shorter than the methane lifetime. We convolve the GEOS-Chem vertical profiles of methane dry mixing ratios with the TROPOMI averaging kernel vectors and prior vertical profiles (*Varon et al*., 2022) to obtain the model simulation of XCH$_4$ for comparison to the TROPOMI observations in the inversion.

Bias in boundary conditions is critical to avoid as it would propagate to biases in the inversion. The boundary condition vertical profiles obtained from *Qu et al.* (2021) avoids systematic drift of the simulation from the TROPOMI observations, but some bias could remain because *Qu et al*. (2021) used an earlier version (1.03) of the TROPOMI data and the data would not be expected to perfectly correct the model anyway. We therefore further correct the boundary conditions on each side of our domain (north, south, west, and east) and for each season as part of the inversion (Table S1). Initial conditions on 1 January 2019 are also from the GEOS-Chem simulations by *Qu et al* (2021) and uniformly scaled to match the mean column mixing ratios retrieved from TROPOMI.

### 2.4. Analytical inversion

The state vector $x$ to be optimized in the inversion includes spatially resolved emissions within the inversion domain and seasonal boundary conditions. We could technically carry out the inversion on the 0.25º × 0.3125º model grid, but satellite observations do not have sufficient information to constrain emissions in such detail everywhere; attempting to do so would introduce large smoothing errors (*Wecht et al*., 2014; *Turner and Jacob*, 2015).Here we use the Gaussian mixture model (GMM) of *Turner* and *Jacob* (2015) to define emission patterns that can be effectively constrained by the TROPOMI observations as informed by the prior estimates. The GMM functions are selected with the goal of retaining native resolution for strong localized source features while merging weak source regions as given by the prior emission field.

Specifically, we project methane emissions at $0.25^\circ \times 0.3125^\circ$ resolution onto $K$-dimensional Gaussian functions where $K$ is the number of similarity criteria, in this case 14 similarity factors on the $0.25^\circ \times 0.3125^\circ$ grid including longitude and latitude (spatial proximity), and the prior emission patterns by sector (Sect. 2.2). Each multivariate Gaussian is hence built to characterize the location (determined by longitude and latitude), emission magnitude and distribution from different sectors (*Turner and Jacob*, 2015). The parameters of the Gaussians are estimated using an expectation-maximization algorithm (*Dempster et al.*, 1977) to find the maximum likelihood. We choose to use 600 Gaussian functions, based on previous experience in inversions for North America (*Turner and Jacob*, 2015; *Maasakkers et al.*, 2021). The inversion optimizes the amplitudes for each Gaussian. We also optimize 16 boundary conditions (four seasons × four

boundaries) for a total of $n = 616$ state vector elements. Construction of the GMM does not include information from the observations and therefore might not resolve hotspots in the observations that are not present in the prior emission patterns. *Nesser et al.* (2021) proposed an alternative approach where information from the observations is integrated into the emission patterns to be optimized.

We perform the inversion with lognormal error probability density functions (pdfs) for prior emissions (*Maasakkers et al.*, 2019; *Lu et al.*, 2022). Specifically we optimize $\ln(\boldsymbol{x})$ instead of $\boldsymbol{x}$, with the prior errors on $\ln(\boldsymbol{x})$ (refer to hereafter as $\boldsymbol{x}'$) following a Gaussian distribution. This enforces positivity of the solution and better captures the high tail of the frequency distribution of emissions than a normal error pdf. High-tailed emissions have been observed for all sectors

including oil/gas (*Yuan et al.,* 2015; *Zavala-Araiza et al.*, 2015; *Lyon et al.*, 2015; *Alvarez et al.*, 2018), coal (*Sadavarte et al.*, 2021), waste (*Maasakkers et al.*, 2022), and livestock(*Duren et al.*, 2019).

Bayesian inference of the maximum a posteriori (MAP) estimate for the state vector $\boldsymbol{x}$ assuming normal error pdfs involves minimization of the cost function $J(\boldsymbol{x})$ (*Brasseur and Jacob*, 2017):

$$J(\boldsymbol{x}') = (\boldsymbol{x}' - \boldsymbol{x}'_a)^\mathrm{T} \mathbf{S}'^{-1}_a (\boldsymbol{x}' - \boldsymbol{x}'_a) + \gamma (\boldsymbol{y} - \mathbf{K}'\boldsymbol{x}')^\mathrm{T} \mathbf{S}^{-1}_o (\boldsymbol{y} - \mathbf{K}'\boldsymbol{x}') \tag{1}$$

where $\boldsymbol{x}' = \ln(\boldsymbol{x})$ and $\boldsymbol{x}'_a = \ln(\boldsymbol{x}_a)$, $\boldsymbol{x}_a$ ($n \times 1$) is the prior emission estimate (Sect. 2.2), and $\boldsymbol{y}$ ($m \times 1$) is the $m$-dimensional vector of TROPOMI observations ($m = 5907939$). $\mathbf{S}'_a$ ($n \times n$) is the prior error covariance matrix in log space and $\mathbf{S}_o$ ($m \times m$) is the observational error covariance

matrix. $\mathbf{K}' = \partial \boldsymbol{y} / \partial \boldsymbol{x}'$ ($m \times n$) is the Jacobian matrix that describes the sensitivity of observations $\boldsymbol{y}$ to $\boldsymbol{x}'$, and $\mathbf{K}'\boldsymbol{x}' = \mathbf{K}\boldsymbol{x}$ where $\mathbf{K} = \partial \boldsymbol{y} / \partial \boldsymbol{x}$ is the sensitivity of $\boldsymbol{y}$ to $\boldsymbol{x}$, which can be readily represented by the GEOS-Chem forward model (*Jacob et al.,* 2016). The relationship between methane emissions and concentrations (or more precisely, concentration enhancement above background mixing ratios) in the nested GEOS-Chem simulation is linear (omitting the minimal effect of potential errors in initial conditions), so that $\mathbf{K}$ defines the forward model for the purpose of the inversion. We hence derive $\mathbf{K}'_{i,j} = \frac{\partial y_i}{\partial \ln(x_j)} = x_j \frac{\partial y_i}{\partial x_j} = x_j \mathbf{K}_{i,j}$, where $i$ and $j$

represent the indices of the observation and the state vector elements. $\gamma$ is a regularization factor to avoid over/underfit to observations. $\gamma$ is needed because the prior and observational error covariance matrices can only be roughly estimated and is assumed here to be diagonal for lack of

better information and convenience of computation.

We assume a geometric standard deviation factor ($\sigma_g$) of 2 for the lognormally distributed errors in $\boldsymbol{x}$ and construct the $\mathbf{S}'_a$ matrix (with diagonal elements $s'_a$) following $\sqrt{s_a'} = \ln(\sigma_g)$ (*Kirkwood*, 1979; *Limpert et al.*, 2001). Observational error standard deviations (square roots of diagonal terms of $\mathbf{S_o}$) include contributions from instrument error, retrieval error, representation error, and forward model error. We calculate the sum of these errors using the residual error method (*Heald et al.*, 2004) on the basis of the XCH4 differences $\Delta = \boldsymbol{y} - \boldsymbol{y_{GEOS-Chem,prior}}$ for individual 0.25°× 0.3125° grid cells between individual TROPOMI observations and the GEOS-Chem simulation with prior emissions. The temporal mean 2019 difference $\overline{\Delta} =$

$\overline{\boldsymbol{y} - \boldsymbol{y_{GEOS-Chem,prior}}}$ for each grid cell is to be corrected in the inversion while the standard deviation of the residual difference $\Delta - \overline{\Delta}$ is taken as the observational error standard deviation, adjusted up to a minimum value of 10 ppb following *Maasakkers et al* (2019) if necessary (10.4% of the retrievals). The resulting observational error standard deviation averages 13.4 ppb, which agrees closely with previous TROPOMI inverse analyses and is mostly due to retrieval error (*Shen et al.*, 2021; *Qu et al.*, 2021). Sparse matrix algebra is applied wherever possible in matrix calculations, making use of the diagonal structure of the error covariance matrices $\mathbf{S}'_a$ and $\mathbf{S_o}$.

As mentioned earlier, the Jacobian matrix $\mathbf{K}'$ is nonlinear and can be immediately transformed following $\mathbf{K}'_{i,j} = x_j \mathbf{K}_{i,j}$. Here we construct $\mathbf{K}$ column by column, by perturbing individual

elements $x_i$ of the state vector independently and running GEOS-Chem forward model simulations to obtain the columns $\partial \boldsymbol{y}/\partial x_i$. These simulations are readily achievable with massively parallel computing.

The optimization problem is nonlinear and needs to be solved iteratively. We approach the solution using the Levenberg-Marquardt method (*Rodgers*, 2000):

$$\boldsymbol{x'}_{N+1} = \boldsymbol{x'}_N + \left(\gamma \mathbf{K}'_N{}^{\mathbf{T}} \mathbf{S}_0'^{-1} \mathbf{K}'_N + (1+k)\mathbf{S}_a'^{-1}\right)^{-1}\left(\gamma \mathbf{K}'_N{}^{\mathbf{T}} \mathbf{S}_0'^{-1}(\boldsymbol{y} - \mathbf{K}\boldsymbol{x}_N) - \mathbf{S}_a'^{-1}(\boldsymbol{x'}_N - \boldsymbol{x'_a})\right) \quad (2)$$

$k$ is a coefficient for the iterative approach, and we tested three methods to set $\kappa$:

(1) $\kappa$ is set to 100 to start and is gradually decreased as the solution is approached, i.e., $\kappa = 101 - \max(N, 101)$ where $N$ is the iteration index;

(2) $\kappa$ is set to 100 for iterations $N \in [1,20)$; 10 for $N \in [21,40)$; 1 for $N \in [41,60)$; and 0 for

$N > 60$;

(3) $\kappa$ is fixed at 10.

We find that using $\kappa = 10$ converges faster with no difference in results compared to the other two methods, and adopt that method in what follows. We iterate on Eq. (2) until the maximum difference in state vector elements between two consecutive iterations ($\boldsymbol{x'_N}$ and $\boldsymbol{x'_{N+1}}$) is smaller than 0.5%, at which point we adopt $\widehat{\boldsymbol{x'}} = \boldsymbol{x'_{N+1}}$ as the best posterior estimate. It takes the base inversion 139 iterations to converge to the solution.

It is of critical importance to discuss if the Gaussian transformation in Eq. (1) could arrive at a best linear unbiased estimator (BLUE) solution (*Cohn*, 1997). As $\boldsymbol{x'} - \boldsymbol{x'_a} \sim N(0, \mathbf{S}'_a)$ and $\boldsymbol{y} - \mathbf{K}'\boldsymbol{x'} = \boldsymbol{y} - \mathbf{K}\boldsymbol{x} \sim N(0, \mathbf{S_o})$ (Fig. S1), both the prior and observational errors are Gaussian with

zero mean; there is a non-linearity relationship between $\boldsymbol{x}'$ and $\boldsymbol{y}$ that are linked by $\mathbf{K}'$. Our analytical transformation thus conforms to the case of a 'Gaussian anamorphosis' defined by *Bocquet et al* (2010), for which a BLUE solution can be properly carried out; a weak point is that the Jacobian matrix may be nonlinear, which is however sometimes the case in particular if the original Jacobian matrix is linear. Previous studies have applied this approach to transform non-Gaussian problems (*Fletcher et al.*, 2010; *Brioude et al.*, 2011; *Saide et al.*, 2015; *Cui et al.*, 2019). We acknowledge that those studies assumed non-Gaussian errors for both the prior information and the observations, while our work only assumes log-normally distributed errors on the prior state vector.

*Rodgers* (2000) indicated that the solution of a non-linear problem using the Levenberg-
Marquardt method can be applied to obtain the posterior error covariance matrix $\widehat{\mathbf{S}}'$:

$$\widehat{\mathbf{S}}' = (\gamma \mathbf{K}'^{\mathbf{T}} \mathbf{S}_{\mathbf{o}}^{-1} \mathbf{K}' + \mathbf{S}_{\mathbf{a}}'^{-1})^{-1} \tag{3}$$

with the averaging kernel matrix $\mathbf{A}$ quantifying the sensitivity of the solution to the true value:

$$\mathbf{A} = \frac{\partial \widehat{x}'}{\partial x'} = \mathbf{I}_{\mathbf{n}} - \widehat{\mathbf{S}}' \mathbf{S}_{\mathbf{a}}'^{-1} \tag{4}$$

where $\mathbf{I}_{\mathbf{n}}$ is the identity matrix. The trace of $\mathbf{A}$ measures the number of independent pieces of information on $\boldsymbol{x}'$ obtained from the observations, and is often referred to as the degrees of freedom for signal (DOFS). The diagonal terms of $\mathbf{A}$ define the averaging kernel sensitivities, which quantify the extent to which the solution is informed by the observations within the inversion framework. They measure the actual error reduction if the inversion framework is correct, but errors in inversion parameters such as prior emission distributions can affect this
interpretation (*Yu et al.*, 2021). An alternative and better way to estimate posterior errors is to generate an ensemble of sensitivity inversions (Sect. 2.6).

The optimal value of $\gamma$ can be determined following *Lu et al.* (2021) so that the sum of state vector terms in the posterior estimate of the cost function, $J_a(\widehat{x}') = (\widehat{x}' - x'_a)^{\mathbf{T}} \mathbf{S}_{\mathbf{a}}'^{-1} (\widehat{x}' - x'_a)$ has value of $\sim n \pm \sqrt{2n}$, which is the expected value ($\pm 1$ standard deviation) from the Chi-square distribution with $n$ degrees of freedom. We find in this manner an optimal $\gamma$ value of 0.015 as the best fit for our TROPOMI inversion. This is smaller than a previous regional TROPOMI inversion at $0.25° \times 0.3125°$ resolution ($\gamma$=0.25 in *Shen et al.*, 2021) because of the much larger number of observations per state vector element ($m/n$) in our work. We also conducted sensitivity inversions using $\gamma$ =0.005 and 0.03 ns as described in Sect. 2.6.

The MAP estimate in log space is for the median of emission but not for the mean; mean emissions are however necessary for spatial aggregation and sectoral attribution purposes. Here we make use of the posterior error covariance matrix from Eq. (3) to infer the mean emissions from the median following the lognormal distribution statistics $x_{mean} = x_{median} \exp\left(\frac{\widehat{s}'}{2}\right)$, and the corresponding analytical posterior error covariance $\widehat{\mathbf{S}}$ (with diagonal elements $\widehat{s} = x_{mean}^2 \exp(\widehat{s}' - 1)$ ), where $\widehat{s}'$ is the diagonal element of the posterior error covariance matrix in log space corresponding to that state vector element. We still use the normal error assumption for the boundary conditions elements of the state vector, with a prior error standard deviation of 10 ppb.

2.5 Sectoral attribution of posterior emissions

The posterior estimate of methane emissions for the GMM state vector can be readily mapped on the 0.25°×0.3125° grid by summation of the GMM elements, but it is also of interest to aggregate it spatially for inferring national totals including by source sector. This reduction in state vector dimension is readily done while preserving the information from the posterior error covariance matrix by using a summation matrix $\mathbf{W}$ to represent the linear transformation from the full state vector ($n \times 1$) to the reduced state vector. Here we use the reduction of the state vector to 12 sectors (Sect. 2.2) of aggregated emissions as an example to illustrate the construction of $\mathbf{W}$. The GMM approach derives the relative weighting of each Gaussian on the $p$ native-resolution grid $\mathbf{W_1}$ ($p \times n$); $\mathbf{W_1}$ thus allows the spatial allocation of posterior state vector to individual 0.25°×0.3125° grid cell (Figs. 4c-d). $\mathbf{W_1}$ is further multiplied by $\mathbf{W_2}$ ($12 \times p$), the fractional

contribution of individual sectors to total grid cell emissions, to obtain the summation matrix $\mathbf{W} = \mathbf{W_2}\mathbf{W_1}$ ($12 \times n$ in this example).

The posterior estimate of the reduced state vector ($\boldsymbol{x_{red}}$) is computed as

$$\hat{\boldsymbol{x}}_{red} = \mathbf{W}\hat{\boldsymbol{x}} \tag{5}$$

and the posterior error covariance and averaging kernel matrices are then given by

$$\hat{\mathbf{S}}_{red} = \mathbf{W}\hat{\mathbf{S}}\mathbf{W}^{\mathbf{T}} \tag{6}$$

$$\mathbf{A}_{red} = \mathbf{W}\mathbf{A}\mathbf{W}^{*} \tag{7}$$

where $\mathbf{W}^{*} = (\mathbf{W}^{\mathbf{T}}\mathbf{W})^{-1}\mathbf{W}^{\mathbf{T}}$ is the Moore-Penrose pseudo inverse (*Calisesi et al.*, 2005).

2.6 Error characterization and inversion ensemble

The sections above describe our base inversion with solution defined by ($\hat{\boldsymbol{x}}, \hat{\mathbf{S}}$). By using the regularization factor $\gamma$, we prevent overfit to the observations and therefore $\hat{\mathbf{S}}$ is a fair representation of the uncertainty within our choice of inversion parameters. However, there is uncertainty in these parameters, and we therefore perform an ensemble of sensitivity inversions with different choices. The sensitivity inversions include (1) using $\ln(1.5)$, and $\ln(2.5)$ for the prior error standard deviations instead of $\ln(2)$; (2) using 0.005 and 0.03 for the regularization factor $\gamma$ instead of 0.015; and (3) using 5 and 20 ppb for the prior error standard deviation in the boundary condition elements of the state vector instead of 10 ppb. We also perform sensitivity inversions assuming normally distributed errors with prior error standard deviation $\sqrt{s_a}$=50%. Combination of these perturbations to our inversion framework generates 36 members in the

inversion ensemble. The uncertainty in posterior estimates reported here is taken as the greater of the range of solutions given by the inversion ensemble and the 2-sigma error inferred from the diagonal of $\hat{\mathbf{S}}$, and is generally determined by the ensemble (Fig. S2).

## 3. Results

3.1 Evaluation of posterior emission estimates

Fig. 4 compares the prior and posterior estimates of emissions mapped on the 0.25º ×0.3125º grid. It also shows the averaging kernel sensitivities (diagonal terms of the averaging kernel matrix), which measure the ability of TROPOMI observations to determine the posterior solution independently of the prior estimate (0=not at all; 1=perfectly). High averaging kernel sensitivities reflect a combination of high observation density and large prior emissions. We achieve high sensitivities to observations in major source regions, with 167 independent pieces of information (DOFS) out of the 600 Gaussian state vector elements.

Comparison of GEOS-Chem simulations using posterior versus prior emissions indicates an improved ability of the posterior emissions to fit the TROPOMI observations (Fig. 1). The mean bias over the inversion domain decreases from 7.8 to 0.4 ppb while the RMSE decreases from 16.8 to 13.6 ppb. The inversion effectively corrects the mean bias from using the prior emissions. The ability to decrease the RMSE is limited by the retrieval error on individual observations.

We independently evaluate the posterior estimate by comparison to *in situ* surface observations from the GLOBALVIEWplus $CH_4$ ObsPack v4.0 database compiled by the National Oceanic and Atmospheric Administration (NOAA) Global Monitoring Laboratory (*Schuldt et al*., 2021). There are five sites in East Asia in 2019, all in relatively remote locations and with near-weekly sampling schedule (Fig. 5 and Table S2). The GEOS-Chem model bias for 2019 annual mean concentrations across the five sites is -4.1 ± 9.5 ppb using prior emissions and -3.8 ± 4.7 ppb using posterior emissions. There is little decrease in the mean bias, which is consistent with the mean bias of -3.4 ppb for TROPOMI relative to TCCON (*Lorente et al.,* 2021) and implies that both the prior and posterior simulations are effectively unbiased in the mean. The factor of 2 lower standard deviation in the posterior simulation indicates a better fit to observations. The RMSE for individual observations decreases only slightly from 23.7 ppb to 20.8 ppb because it is limited by the forward model transport error, previously estimated by *Lu et al.,* (2021) at 20 ppb for the GLOBALVIEWplus $CH_4$ ObsPack database using the residual error method. The model transport error is larger for surface than satellite observations because the amplitude of variability is larger and includes uncertainties in boundary layer vertical mixing.

*Qu et al* (2021) previously reported overcorrections and inconsistencies with respect to GOSAT in their global TROPOMI inversion results over southeastern China. They attributed the problem to spatial overlap of coal and rice emissions, and to seasonal cloudiness correlated with the peak in rice emissions. We have more confidence in our results for several reasons. First, our higher spatial resolution compared to the 2º×2.5º of *Qu et al* (2021) allows better separation of coal and rice emissions. Second, we use an improved spatial distribution of coal emissions (*Sheng et al*., 2019) compared to the EDGAR v4.3.2 inventory in *Qu et al*. (2021). Third, we use version 2.02 of the TROPOMI retrieval with additional filters, and exclude data inconsistent with GOSAT (Fig. 2), whereas *Qu et al.* (2021) used TROPOMI v1.03 data with quality flags but no other filtering. Our results show higher averaging kernel sensitivities over southeastern China than *Qu at al*. (2021) and no overcorrections (Fig. 1).

3.2 National and sectoral emissions for China

Table 1 compiles the total national and sectoral posterior emissions for China. Sectoral attribution assumes that the posterior/prior emission ratios for a given 0.25º×0.3125º grid cell (Fig. 4c) apply equally to all prior emission sectors within that grid cell, so that the combination of Fig. 4c and Fig. 3 gives the spatial distribution of the change in emission by sector. Posterior estimates of total, anthropogenic, and natural emissions for China are 70.0 (61.6-79.9), 65.0

(57.7-68.4), and 5.0 (3.9-11.6) Tg a$^{-1}$, respectively, where the parentheses indicate the uncertainty range in the inversion solution as described in Sect. 2.6. The averaging kernel sensitivities for the national total and anthropogenic posterior emission estimates are 0.91, indicating that these estimates are largely determined by the TROPOMI observations with little influence from the prior estimate. Our best posterior estimate of 65 Tg a$^{-1}$ for Chinese anthropogenic emissions is 21% higher than the 2014 value of 53.6 Tg a$^{-1}$ reported by the Chinese government to the UNFCCC, and the range of our inversion results gives us high confidence that the reported emissions are too low.

Our ability to separate the contributions from different sectors to the posterior emission estimates for China can be evaluated by examining the error correlations in the reduced posterior error
covariance matrix (Sect. 2.5), as shown in Fig. 6. We find that landfills and wastewater treatment cannot be effectively separated in the posterior solution (posterior error correlation coefficient $r$ = 0.95), because they have similar spatial distributions associated with population (Fig. 3), and we thus group them as a single waste sector for further analysis. The 'Other' sector, which is mostly associated with urban emissions, also has strong error correlations with waste ($r$ = 0.66-0.75). Other sectors can be successfully separated, as shown by the posterior error correlations in Fig. 6. We find that most of the posterior error correlation coefficients between sectors are lower than 0.2. For example, there is little error correlation ($r$ = -0.2 to 0.1) between coal and other sectors. The global TROPOMI inversion by *Qu et al* (2021) found it difficult to separate emissions between coal and rice paddies, but here we find a low error correlation of -0.04 that
reflects our much higher spatial resolution. The main natural emission sector is wetlands, which is effectively separated from all other sectors except rice ($r$ = 0.29).

We can now attribute the 21% underestimate of anthropogenic emissions in the Chinese government report to the UNFCCC, as given in Table 1. We find large upward corrections in emissions from oil (+147%), gas (+61%), livestock (+37%), rice paddies (+34%), and waste (+41%), but a downward correction in coal emissions (-15%). Averaging kernel sensitivities for all anthropogenic sectors are high (0.71-0.91), indicating strong constraints from the observations. An exception is the gas sector (0.31), for which emissions are relatively small. The uncertainty ranges of the inversion results for the different sectors are small, indicating an insensitivity to different inversion assumptions and high confidence in the posterior sectoral
emissions in the base inversion.

The inversion returns a larger estimate of 5.0 (3.9-11.6) Tg a$^{-1}$ for natural sources relative to 3.2 Tg a$^{-1}$ in the prior estimate, mainly driven by increased contributions from wetlands (+1.1 Tg a$^{-1}$) and termites (+0.5 Tg a$^{-1}$). Averaging kernel sensitivity for wetlands is moderately high (0.61) but low for the other small natural sources.

The base inversion assuming a lognormal error distribution for prior emissions returns larger posterior Chinese emissions from all sectors relative to a normal error assumption, as would be expected from the asymmetry of the lognormal function. The largest differences are for the oil and gas sectors, where the sensitivity inversion assuming a normal error distribution yields posterior estimates respectively 22% and 21% lower than the base inversion. The oil and gas sectors are
particularly high-tailed in their frequency distributions of emissions (*Zavala-Araiza et al*., 2015; *Lyon et al*., 2015; *Brandt et al.,* 2016; *Alvarez et al*., 2018).

## 4. Discussion

By using the official Chinese inventory reported to the UNFCCC as prior estimate of methane emissions, our inversion can usefully evaluate that inventory and guide its improvement. Here we discuss the significance and implications of our results for different sectors, placing them in the context of previous literature, and we identify specific issues requiring further work.

Fig. 7 compiles the total and sectoral anthropogenic emissions in China reported by top-down and bottom-up studies for the past decade. Our total posterior emission estimate of 65 (57.7-68.4) Tg a$^{-1}$ is consistent with the EDGAR inventories but this reflects canceling differences for individual sectors as shown in the bottom panel. We estimate higher national total emissions than *Peng et al* (2016), driven by their much smaller rice and waste emissions. Our estimate is higher than the best estimates from previous top-down studies (43-62 Tg a$^{-1}$), which used EDGAR prior estimates for spatial distribution, and were conducted at much coarser resolutions (2°×2.5°-4°×5° versus 0.25°×0.3125°) and with much sparser observations (GOSAT versus TROPOMI) than ours. *Deng et al* (2022) compiled results from 11 GOSAT inversions for 2010-2017 using Chinese UNFCCC totals as prior estimate and showing a range of 40-62 Tg a$^{-1}$. Although the emission estimates in Fig. 7 are from different years, *Lu et al*. (2021) and *Y. Zhang et al*. (2021) reported an increasing Chinese emission trend of 0.4 Tg a$^{-1}$ from inversion of 2010-2018 GOSAT observations, which would only make a small contribution to the differences. The lower emissions in the previous top-down studies are mostly driven by downward revision of coal emissions relative to their prior estimates, and we find such a decrease too but not to the same extent. Our inversion uses an improved prior estimate of the distribution of coal emissions in China with much larger contribution from southern China (*Sheng et al*., 2019), so that some of the previous corrections attributed to rice agriculture might reflect coal emissions instead. Another striking difference in our work relative to others is the much higher livestock emissions. We discuss the coal, oil/gas, livestock, and waste sectors in more detail below.

### 4.1 Coal

Our downward correction of coal emissions compared to the UNFCCC report is driven by both Shanxi province and southwestern China, which are the two largest coal producing regions in China (*Zhu et al*., 2017). This could reflect (1) overestimate of EFs in the UNFCCC report, (2) under-accounting of surface mines, and (3) increasing coal methane utilization. With regard to (1), many high methane-content coal mines with inefficient coal production have been closed in the past decade (*Wang et al*., 2020). Coal production has shifted from southern and eastern China to northwestern China (including Shanxi) with abundant coal reserves and where methane content is relatively low (*Gao et al*., 2021; *Liu et al*., 2021). With regard to (2), a previous study (*Gao et al*., 2021) indicated an underestimated share of surface mining in the UNFCCC report for China. The methane emission intensity of surface mining is ten times lower than underground mining (*Palmer et al*., 2021). With regard to (3), coal mine methane (CMM) utilization in China has greatly increased in the past decade (*Y. Lu et al*., 2021).

However, uncertainty in the spatial distribution of coal emissions remains a major obstacle for top-down studies. Different bottom-up inventories are inconsistent in their estimates of the number of mines in China (e.g., 324 in EDGAR v4.2, 4243 in EDGAR v4.3.2, and 10963 in *Sheng et al*., 2019). Mine closures and regional shifts in coal production may also be difficult to track (*Gao et al*., 2021). New satellite observations of methane plumes from individual point sources could provide important new information for geolocation and quantification of emissions

from coal mines, as shown for the Shanxi province by *Guanter et al*. (2021) and *Sánchez-García et al.* (2022).

4.2 Oil/gas

Our posterior estimate of oil/gas emissions is higher than the UNFCCC report and *Peng et al* (2016), but lower than EDGAR v4.3.2 and v6. The previous top-down estimates in Fig. 7 range from 0.7 Tg a$^{-1}$ by *Lu et al* (2021) to 5.5 Tg a$^{-1}$ by *Miller et al* (2019), and our estimate is in mid-range. *Scarpelli et al* (2022) found that the oil/gas emissions from *Lu et al* (2021) were heavily influenced by the low GFEI v1 inventory (*Scarpelli et al*., 2020b) used as their prior estimate. The high emissions in *Miller et al* (2019) could reflect their use of the EDGAR v4.2 inventory as prior estimate with spuriously high emissions from pipelines (*Scarpelli et al*., 2020b).

We find that the oil sector has the largest relative upward correction to the UNFCCC inventory (+147%) among all sectors. The correction might be attributed to methane leakage from oil extraction not fully accounted for in the UNFCCC report (*Rutherford et al*., 2021; *Deng et al*., 2022). *Lauvaux et al* (2022) used 2019-2020 TROPOMI observations to identify a number of ultra-emitters (>25 tons h$^{-1}$ on the 5.5×7 km$^2$ grid) from oil production fields; their identified ultra-emitters in China are consistent with the locations where we find large oil upward adjustments.

Gas emissions for China in the UNFCCC report (0.18 Tg a$^{-1}$) are dominated by distribution (0.125 Tg a$^{-1}$) with only small contributions from production (0.03 Tg a$^{-1}$) and transmission (0.025 Tg a$^{-1}$). This is in part because of low EFs from production, and in part because a large fraction of the gas used in China is imported. The assumed EF for gas production in the UNFCCC report is $1.3\times10^{-10}$ Gg per m$^3$ of gas production, much lower than in the IPCC (2006) EF guidelines (lower-end value of $3.8\times10^{-10}$ Gg per m$^3$ of gas production for developed countries). 42% of the gas used in China in 2019 was imported (EIA, 2020).

Our inversion returns a posterior emission for the gas sector of 0.29 Tg a$^{-1}$ including 0.16 Tg a$^{-1}$ from distribution, 0.07 Tg a$^{-1}$ from production, and 0.06 Tg a$^{-1}$ from transmission. However, the averaging kernel sensitivity is low (0.3) and the distribution subsector is difficult to disentangle from the waste sector because it is mostly urban. *Alvarez et al* (2012) suggested that the life-cycle loss rate from gas production should be less than 3.2% for a coal-to-gas transition to be of climate benefit. Our posterior estimate indicates a loss rate of 1.7 (1.3-1.9) % for China, assuming 92% methane gas by volume (*Scarpelli et al*., 2022).

China's gas industry has entered a rapid development stage driven by the domestic coal-to-gas transition policy (*Qin et al*., 2018); China's gas production in 2019 was 42% higher than in the 2014 year of the UNFCCC inventory (*EIA*, 2020). The small loss rate suggests that the transition will be beneficial for climate, but is somewhat misleading because of the large fraction of imported gas. 25% of that imported gas is from Turkmenistan (*EIA*, 2020), where emissions from gas production are exceedingly high (*Varon et al*., 2019; *Ikakulis-Loixalte et al*., 2022; *Lauvaux et al*., 2022). A more complete accounting of the loss rate in China from gas production would factor in the effect of international trade.

4.3 Livestock

Our estimate of Chinese livestock emissions is higher than any previous study (Fig. 7). This is because our inversion corrects livestock emissions upward in northwestern and northeastern

China, where existing bottom-up inventories show weak emissions (*Lin et al.*, 2021). Previous
GOSAT inversions had poor observational coverage over these regions (Fig. 5 in *Qu et al.*,
2021), and their inversion solutions hence cannot depart sufficiently from the low bottom-up
inventories used as prior estimates. However, TROPOMI observations provide strong constraints
as illustrated by the high averaging kernel values (Fig. 3).

4.4 Waste

We estimate higher waste emissions than the 2010 *Peng et al* (2016) inventory and the 2014
UNFCCC report, and we attribute this in part to the rapid development of wastewater treatment
in China. China has enacted major policies on water pollution prevention since 2014 (*Xu et al.*,
2020), including new standards for sewage discharge (*Han et al.*, 2016). The number of
wastewater treatment plants increased by 44% from 2014 to 2019 (*Xu et al.*, 2020). Solid waste
generation has also increased in the past decade (*Sheng et al.*, 2021) but an increasing fraction is
incinerated rather than landfilled (*Liu et al.*, 2021).

**5. Conclusions**

We estimated 2019 methane emissions in China by high-resolution inversion of TROPOMI
satellite observations. Our inversion uses as prior estimate the Chinese national inventory
reported to the UNFCCC so that our results are directly relevant for evaluating and improving
that inventory.

Our inversion uses an analytical solution to the Bayesian inference of methane emissions from
the TROPOMI observations, providing closed-form statistics on posterior errors and information
content as part of the solution. It optimizes a 600-member Gaussian mixture model (GMM) of
emissions, in which concentrated source regions are quantified at up to $0.25^{\circ} \times 0.3125^{\circ}$ ($\approx 25 \times 25$
km$^2$) resolution while regions with weak emissions are spatially aggregated. We assume
lognormal error distributions for the prior emissions, which allows better representation of the
high-tailed component. The Jacobian sensitivity matrix constructed for our analytical solution
enables immediate generation of an inversion ensemble to explore the dependence of the solution
on uncertainties in inversion parameters. This ensemble is combined with the posterior error
covariance matrix of the base inversion to provide a conservative estimate of errors on inferred
emissions including from different sectors. Independent evaluation of inversion results with
surface sites from the NOAA GLOBALVIEWplus CH4 ObsPack v4.0 database shows significant
improvement in the ability to fit the observations.

We estimate from the inversion a total emission for China of 70.0 (61.6-79.9) Tg a$^{-1}$, where the
parentheses indicate the uncertainty ranges. Total anthropogenic emission for China is 65.0
(57.7–68.4) Tg a$^{-1}$ including 16.6 (15.6-17.6) Tg a$^{-1}$ from coal, 2.3 (1.8-2.5) Tg a$^{-1}$ from oil, 0.29
(0.23-0.32) Tg a$^{-1}$ from gas, 17.8 (15.1-21.0) Tg a$^{-1}$ from livestock, 9.3 (8.2-9.9) Tg a$^{-1}$ from
waste, 11.9 (10.7-12.7) Tg a$^{-1}$ from rice paddies, and 6.7 (5.8-7.1) Tg a$^{-1}$ from other sources. Our
inferred total anthropogenic emission for China is 21% higher than the national inventory
reported by the Chinese government to the UNFCCC (53.6 Tg a$^{-1}$). This reflects upward
corrections to emissions from oil (+147%), gas (+61%), livestock (+37%), waste (+41%), and
rice paddies (+34%), and a downward correction in coal emissions (-15%).

Our estimate of anthropogenic Chinese emissions is at the high end of the range of past inversion studies (43-62 Tg a$^{-1}$) that used the much sparser GOSAT satellite observations, coarser resolution, and versions of the EDGAR inventory as prior estimates. We find in particular higher emissions from coal, livestock, and waste. The higher emission from coal may reflect our improved accounting of sources in southern China (*Sheng et al.*, 2019) and a higher spatial resolution that allows us to better separate emissions from coal and rice paddies (*Qu et al.*, 2021). Our upward correction to livestock emissions is mostly in northwestern and northeastern China, where TROPOMI provides much denser information than was previously achievable from GOSAT. Our high estimate of emissions from waste may be driven in part by the large increase in wastewater treatment plants in China over the past decade.

Our upward corrections relative to the Chinese government inventory are largest for oil and gas, even though the contributions from these two sectors to total national methane emissions are still small compared to other sectors. We find high emissions from oil production in the same locations where *Lauvaux et al.* (2022) identified 'ultra-emitters' in the TROPOMI data, suggesting that much of these emissions originate from malfunctioning or poorly operated equipment. Most of the gas emissions in China are from distribution, reflecting low emission factors from gas production but also a large share of imported gas. Emission from gas may increase in the future as China undergoes a coal-to-gas transition in energy policy (*Qin et al.*, 2018) with increasing domestic gas production. We derive a life-cycle loss rate of 1.7 (1.3-1.9) % from gas production in China, lower than the 3.2% break-even point for a coal-to-gas
transition to be beneficial for climate (*Alvarez et al.*, 2012). However, this does not account for imported gas from countries such as Turkmenistan where emission per unit of gas production is exceedingly high.

        *Acknowledgments*. This work was funded by the Climate and Clean Air Coalition (CCAC) of the United Nations Environment Programme (UNEP) and by the NASA Carbon Monitoring System, and by the Harvard University Climate Change Solutions Fund (CCSF).

*Data availability*. The TROPOMI satellite observations version 2.02 are available at *http://www.tropomi.eu/data-products/methane*; The GOSAT methane retrievals version 9.0 are available at *https://catalogue.ceda.ac.uk/uuid/18ef8247f52a4cb6a14013f8235cc1eb*;The ObsPack GLOBALVIEWplus CH4 ObsPack v4.0 data product is available at *https://gml.noaa.gov/ccgg/obspack/data.php*.

        *Author contributions*. ZC and DJJ designed the study. ZC conducted the data and modeling analysis with contributions from HN, MPS, AL, DJV, XL, LS, ZQ, EP, and XYY. ZC and DJJ wrote the paper with input from all authors.

        *Competing interests*. The authors declare that they have no conflict of interest.

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

980

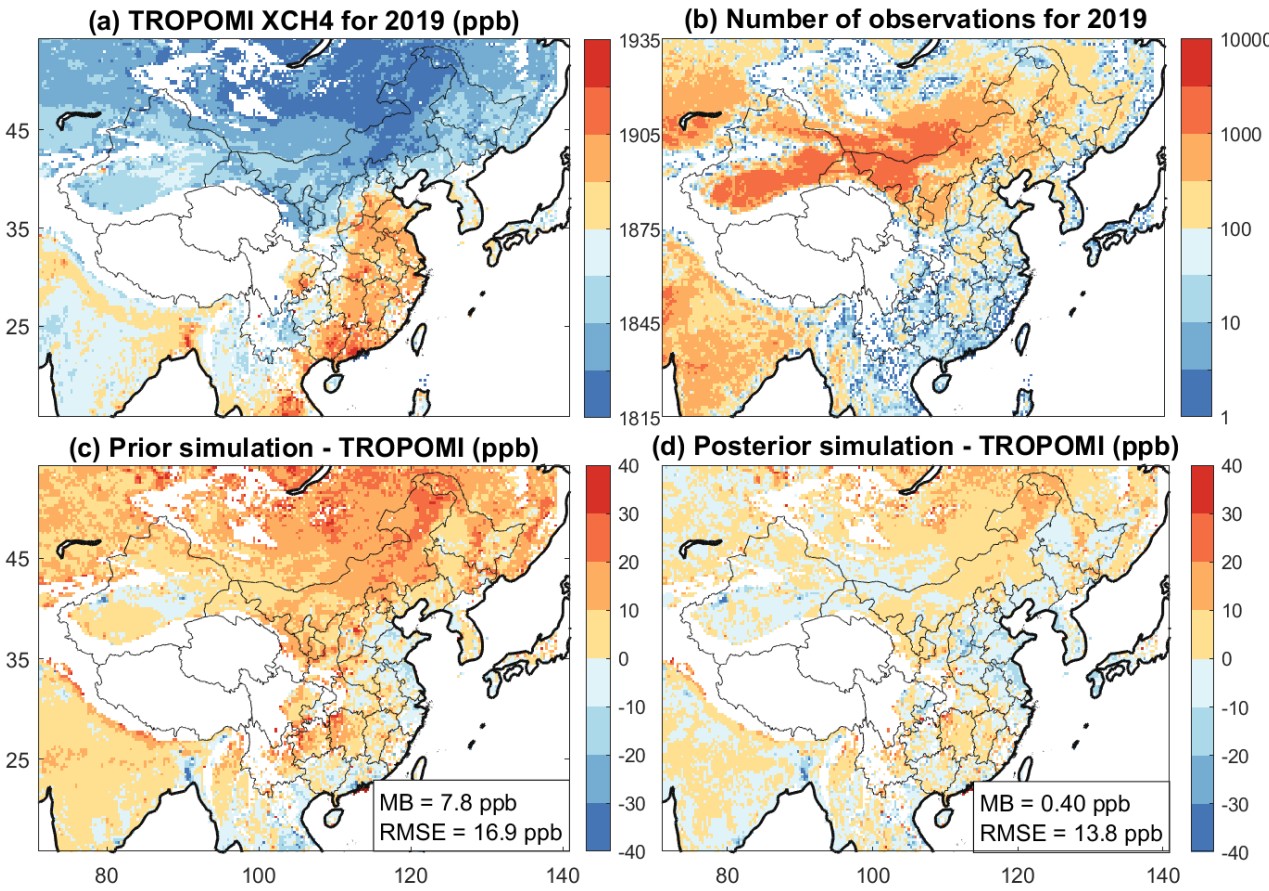

**Figure 1**. TROPOMI observations of column-averaged dry methane mixing ratios (XCH$_4$) over East Asia and comparison to GEOS-Chem simulations. (a) Mean observations for 2019 mapped on the GEOS-Chem 0.25º × 0.3125º grid. (b) Number of observations on that grid. (c) Mean differences between the GEOS-Chem simulation with prior emissions and observations. The spatiotemporal mean bias (MB) and root-mean-square error (RMSE) over the study domain are shown inset. (d) Same as (c) but for the GEOS-Chem simulation with posterior emissions. Thin black lines are Chinese provincial boundaries.

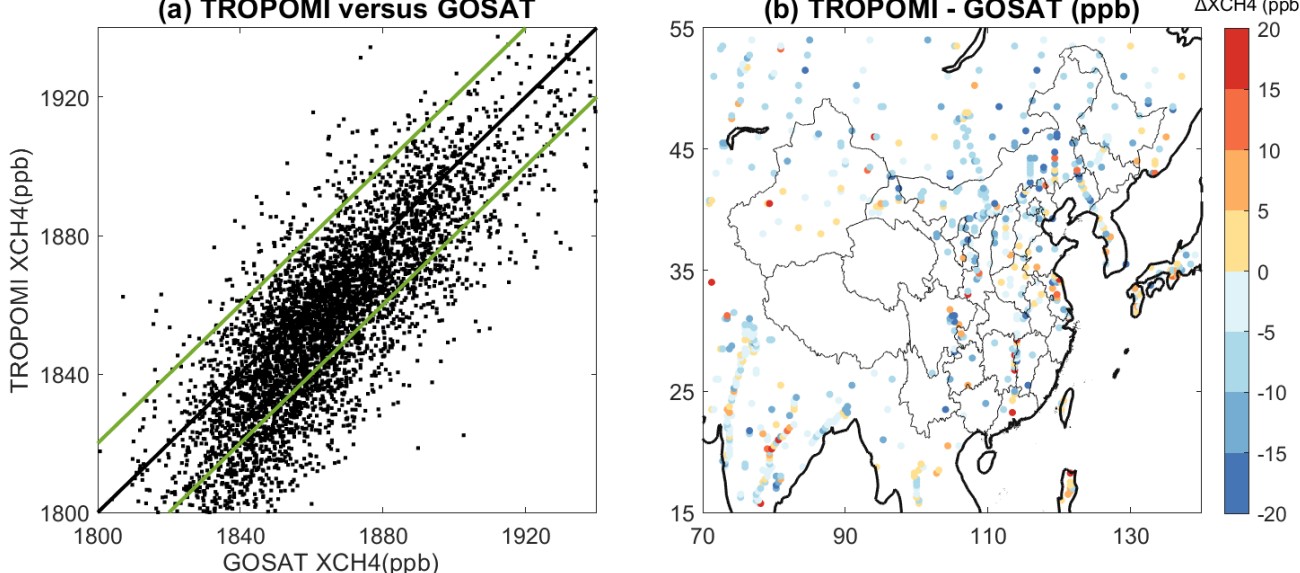

**Figure 2**. Comparison between 2019 TROPOMI and GOSAT observations of XCH4 over East Asia. (a) Scatter plot of daily observations on the GEOS-Chem 0.25º ×0.3125º grid. Green lines indicate absolute TROPOMI-GOSAT differences of 20 ppb and we exclude the outlying TROPOMI observations. (b) Spatial pattern of annual mean differences ΔXCH4 between TROPOMI and GOSAT observations after outlying TROPOMI data have been excluded. The mean difference is -3.6 ± 9.1 ppb. TROPOMI version 2.02 observations are from *Lorente et al.* (2021) and GOSAT version 9.0 observations are from *Parker et al.* (2020).

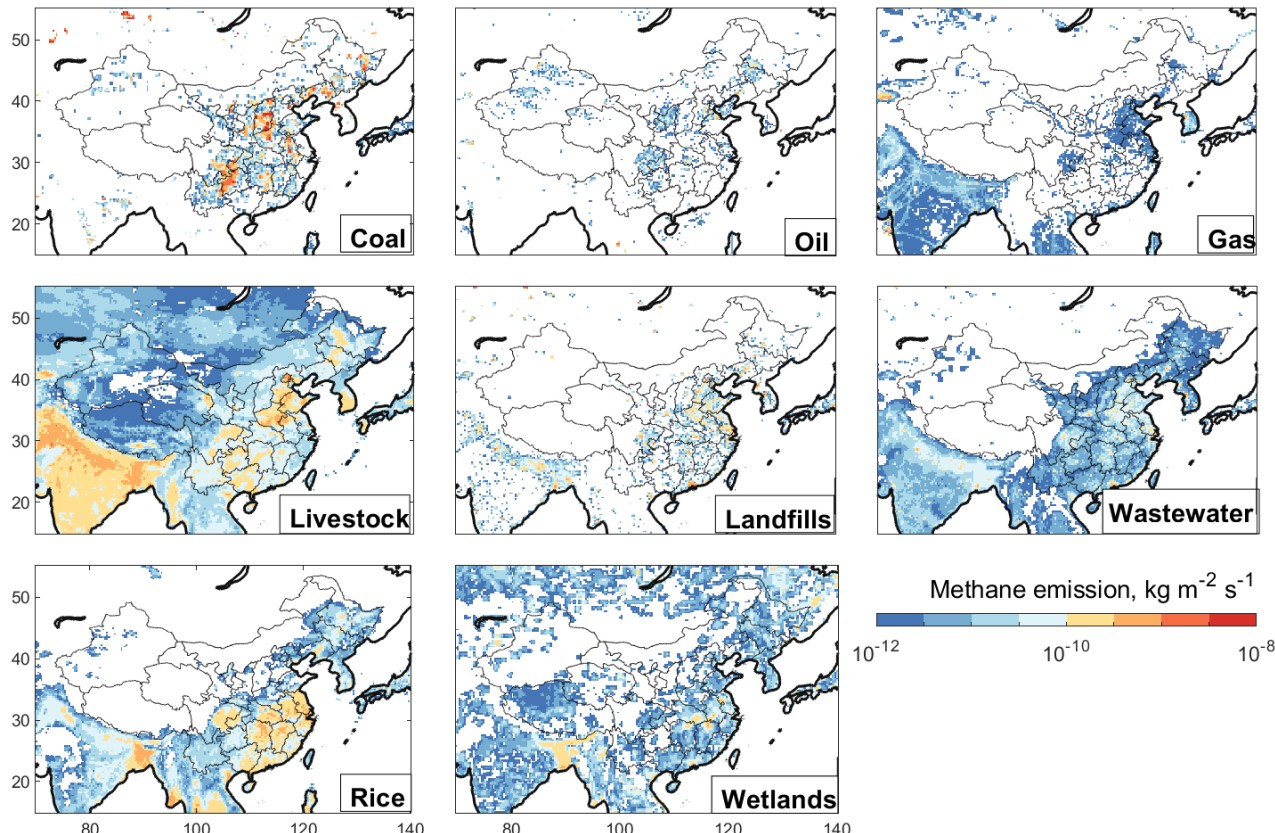

**Figure 3**. Prior estimates of methane emissions used for the inversion. Coal, oil, and gas emissions are from the GFEIv2 gridded version of the national inventories from individual countries reported to the UNFCCC (*Scarpelli et al*., 2022). Other anthropogenic emissions for China are from its UNFCCC report with spatial allocation from EDGAR v4.3.2, while for other countries they are from EDGAR v4.3.2. Wetland emissions are 2019 monthly means of the nine-member high-performance subset of the WetCHARTs inventory ensemble (*Ma et al*., 2021) and are shown here as the annual means for 2019. White areas have emissions lower than $1 \times 10^{-12}$ kg $m^{-2}$ $s^{-1}$. Total emissions for China are listed in Table 1.

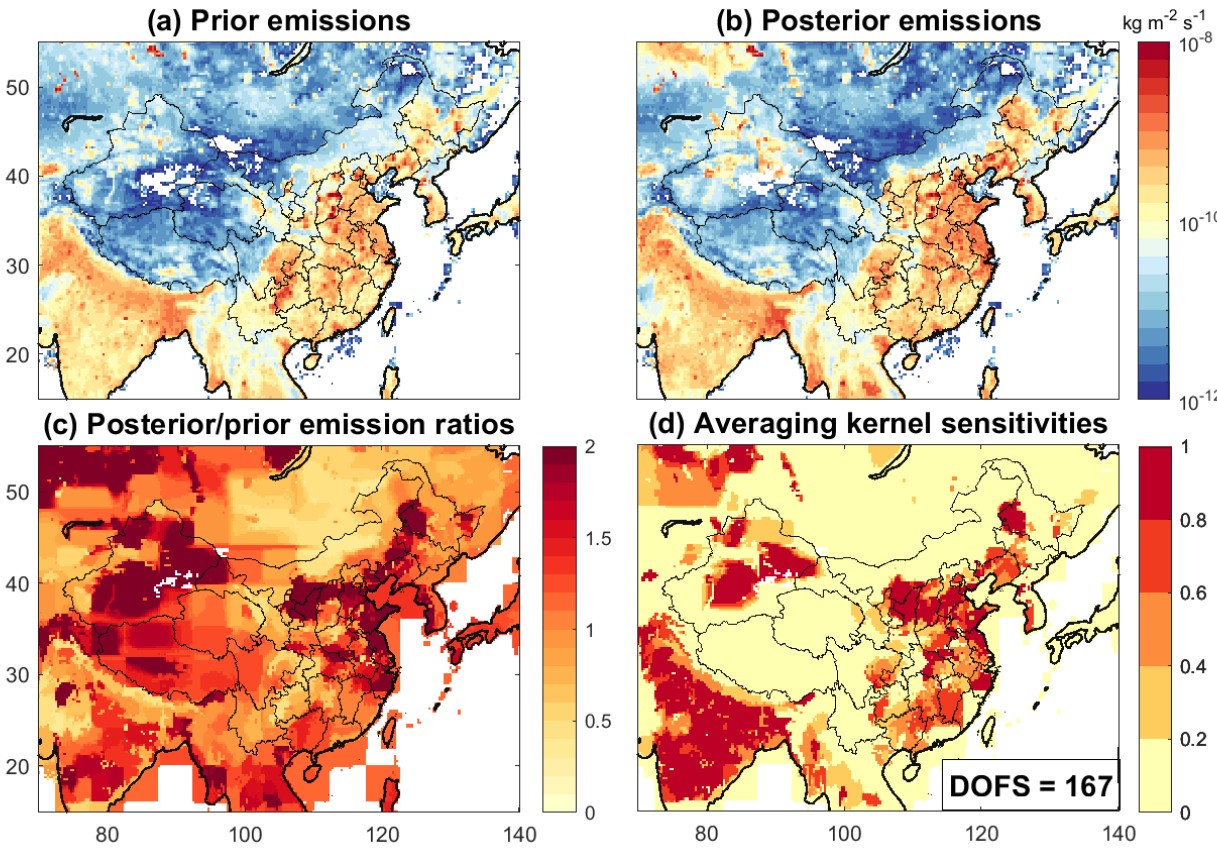

**Figure 4**. Optimization of methane emissions over East Asia in 2019 from inversion of TROPOMI observations. Results are from the base inversion and are shown on the 0.25°× 0.3125° grid. (a) Prior estimates of methane emissions, summing the contributions from the sectors in Fig. 3 plus additional minor sectors as given in Table 1. (b) Posterior methane emissions from the TROPOMI inversion. (c) Posterior/prior emission ratios. (d) Averaging kernel sensitivities. The averaging kernel sensitivities are the diagonal elements of the averaging kernel matrix and display the ability of the observations to quantify emissions independently from the prior estimates (1 = fully, 0 = not at all). The degrees of freedom for signal (DOFS, defined as the trace of the averaging kernel matrix) is given inset.

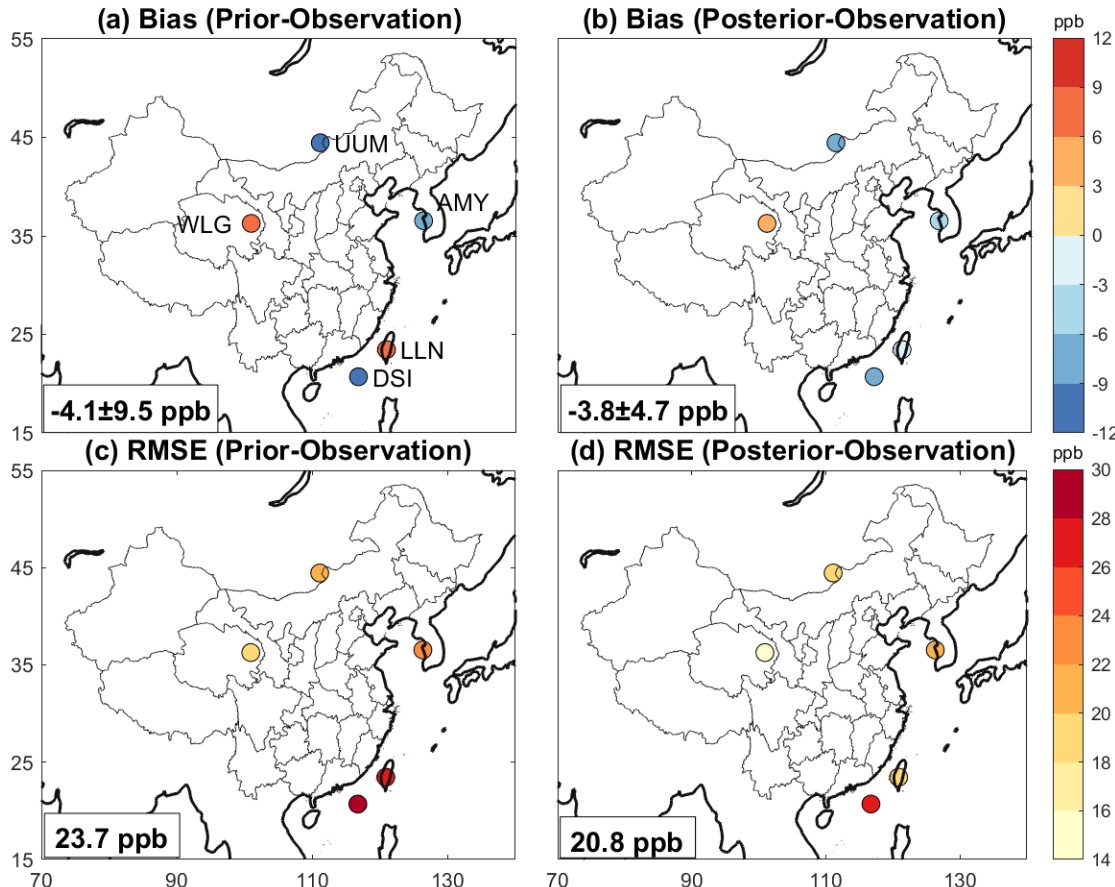

**Figure 5**. Comparison of GEOS-Chem simulations of atmospheric methane concentrations to *in situ* observations from five surface sites in 2019 compiled in the NOAA GLOBALVIEWplus CH₄ ObsPack v4.0 database. The five sites are described in detail in Table S2. The annual mean GEOS-Chem model biases and root-mean-square errors (RMSEs) for individual near-weekly observations at each site are shown. The insets give spatial mean biases ± standard deviations for the ensemble of sites and corresponding RMSEs.

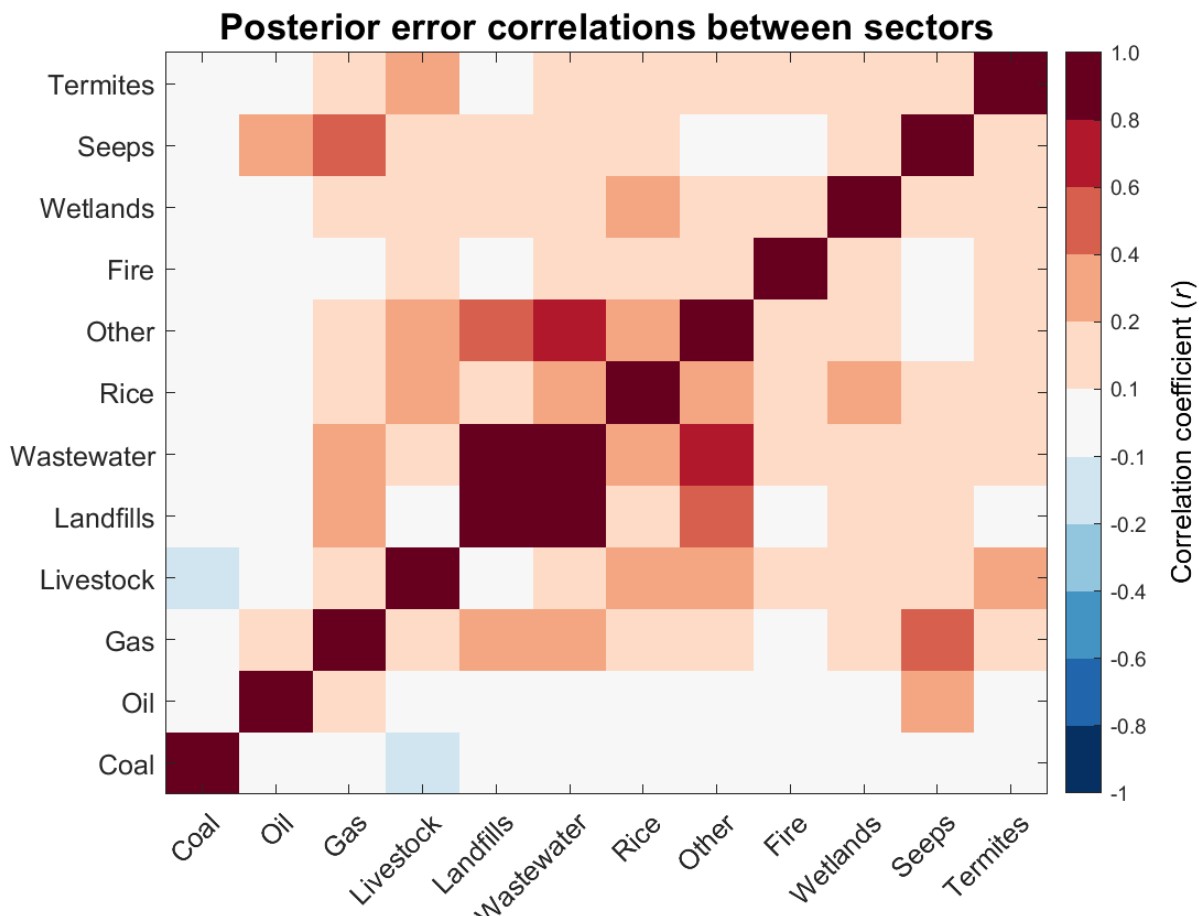

**Figure 6**. Error correlation coefficients (*r*) between posterior estimates of methane emissions from different source sectors in China. Error correlations measure the ability of the inversion to separate emissions between sectors (±1: not at all; 0: fully). 'Other' is a combination of minor anthropogenic emissions including industry, stationary combustion, mobile combustion, aircraft, composting, and field burning of agricultural residues.

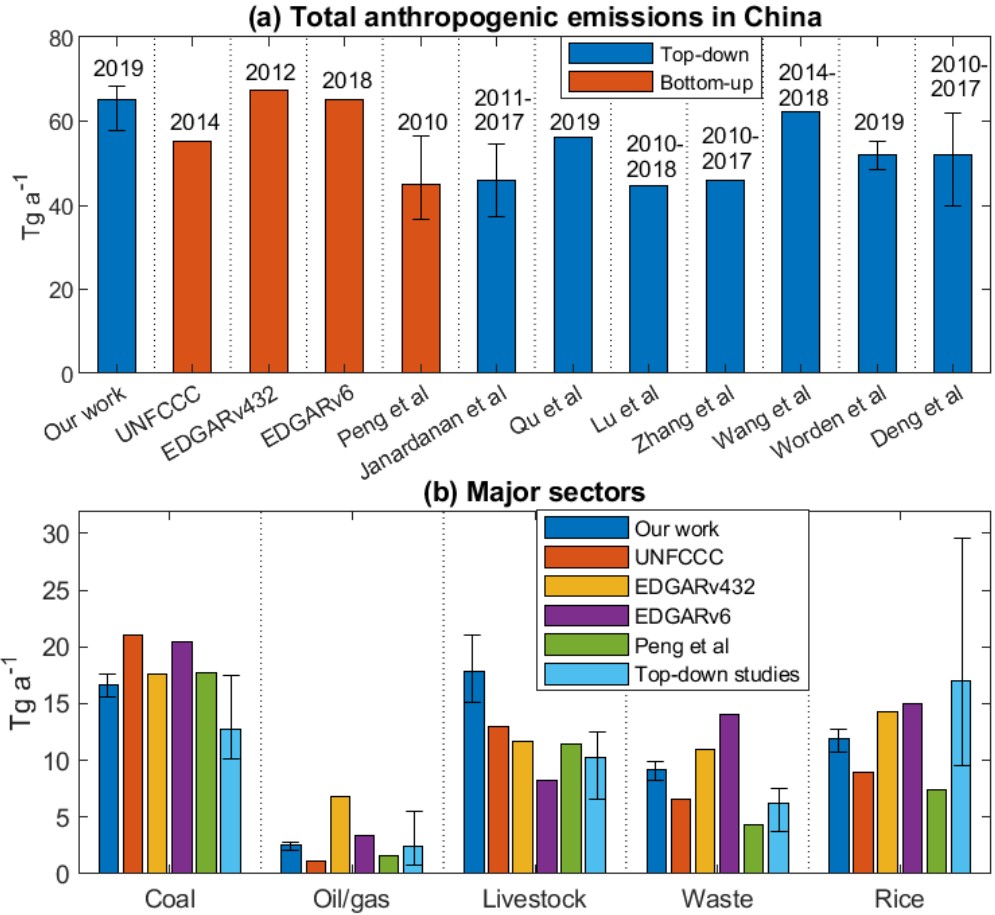

**Figure 7**. Anthropogenic methane emissions in China. The top panel compares the national total emissions reported by different bottom-up and top-down studies. Vertical bars for our work indicate the uncertainty range, obtained by combining results from the inversion ensemble and the posterior error covariance $\hat{\mathbf{S}}$ (Fig. S2). *Peng et al* (2016), *Janardanan et al* (2020), *Worden et al* (2022), and *Deng et al* (2022) also reported uncertainty estimates. The bottom panel shows the contributions from different sectors inferred in our work and compared to inventories including the UNFCCC, EDGAR, and *Peng et al* (2016), and to the means and ranges from recent top-down studies compiled in Table S3.

**Table 1**. Methane emissions in China in 2019.

| | Prior estimate (Tg a$^{-1}$)[a] | Posterior estimate (Tg a$^{-1}$)[b] | Sensitivity to observations[c] |
|---|---|---|---|
| **Total emission** | 56.8 | 70.0 (61.6-79.9) | 0.91 |
| **Anthropogenic** | 53.6 | 65.0 (57.7-68.4) | 0.91 |
| Coal mining | 19.5 | 16.6 (15.6-17.6) | 0.91 |
| Oil | 0.93 | 2.3 (1.8-2.5) | 0.76 |
| Gas[d] | 0.18 | 0.29 (0.23-0.32) | 0.30 |
| Livestock[e] | 13.0 | 17.8 (15.1-21.0) | 0.75 |
| Waste[f] | 6.6 | 9.3 (8.2-9.9) | 0.71 |
| Rice paddies | 8.9 | 11.9 (10.7-12.7) | 0.86 |
| Other[g] | 4.6 | 6.7 (5.8-7.1) | 0.81 |
| **Natural** | 3.2 | 5.0 (3.9-11.6) | 0.61 |
| Open fires[h] | 0.16 | 0.24 (0.18-0.26) | 0.30 |
| Wetlands | 2.3 | 3.4 (2.8-7.5) | 0.61 |
| Seeps | 0.06 | 0.11 (0.10-0.18) | 0.43 |
| Termites | 0.72 | 1.2 (0.8-3.7) | 0.32 |

[a]Prior estimates of anthropogenic emissions are from the Chinese government report to the UNFCCC for 2014 (*UNFCCC*, 2020). Wetland emissions are the mean of the high-performance subset of the WetCHARTs v1.3.1 inventory ensemble for 2019 (*Ma et al*, 2021). Open-fire emissions are from GFED4s (*van der Werf et al*., 2017). Termite emissions are from *Fung et al* (1991), and geological seepage emissions are from *Etiope et al* (2019) with scaling from *Hmiel et al*. (2020). See Sect. 2.2 for details.

[b]Results from the base inversion, with uncertainty range in parentheses encompassing the best estimates from the inversion ensemble and the 2-sigma error from the posterior error covariance matrix **Ŝ** of the base inversion (Fig. S2). See Sect. 2.6 for details.

[c]Sensitivity of posterior emissions to the TROPOMI observations, ranging from 0 (no information from observations, emissions determined by prior estimate) to 1 (full information from observations, no sensitivity to prior estimate). The sensitivities are defined by the diagonal terms of the reduced averaging kernel matrix for the inversion. See Sect. 2.5 for details.

[d]Contributions from production, transmission, and distribution sub-sectors are 0.03, 0.025, and 0.125 Tg a$^{-1}$ in the prior estimate, and are 0.07, 0.06, and 0.16 Tg a$^{-1}$ in the posterior estimate.

[e]Livestock sector includes emissions from enteric fermentation and manure management.

[f]Waste sector includes emissions from landfills and wastewater treatment, and are combined in the inversion because of their spatial overlap. Prior estimates are 3.84 Tg a$^{-1}$ for landfills and 2.72 Tg a$^{-1}$ for wastewater treatment.

[g]Including industry, stationary combustion, mobile combustion, aircraft, composting, and field burning of agricultural residues.

[h]Excluding field burning of agricultural residues.