# Peer review of "Methane emissions from China: a high-resolution inversion of TROPOMI satellite observations"

_Atmospheric Chemistry and Physics, 2022_

## Author Comment (AC1)

We thank the reviewers for their detailed suggestions and comments on the manuscript. Below, we have replied to each review and have detailed the corresponding edits that we have made to the manuscript. We have listed out the reviewer comments in *black italic* and the replies in blue.

**RC1: Dr. Daven Henze**

*The manuscript by Chen et al. presents valuable and timely estimates of sector-specific CH4 emissions in China inferred from TROPOMI data using an inversion with the GEOS-Chem model. Overall, the topic, scope, and presentation are well suited for ACP, and the results are of value to the community. The paper emphasizes the importance of using log-normal distributions to describe CH4 emissions, and so I focused on this a bit in my review. In this regard, though, I'm concerned about one issue described in detail below. Given the importance of this part of their approach, as it related to identification of sector-specific emissions, I think it warrants thoughtful revisions. I've provided comments on some other points as well, again focusing mostly on aspects of the inversion method.*

We thank the reviewer for the insightful comments. We have revised the manuscript following the reviewer's specific suggestions as below. In particular we have re-constructed Sect. 2.4, to more clearly elaborate on the analytical method for an inversion problem with lognormally distributed prior errors.

*Major comments:*

*I do have a question about the overall framework as presented here, and in prior work from this group applying the same approach. To start with, I would appreciate seeing what equation is being minimized by Equation 5. In other words, can the cost function for the non-Gaussian case be written out, explicitly? There is a bit of literature on this topic, and results one could adopt readily, starting from this point, but without seeing what cost function they are actually minimizing, I'm inferring a bit.*

We have written out the cost function on page 6 lines 251-261:

'Bayesian inference of the maximum a posteriori (MAP) estimate for the state vector $x$ assuming normal error pdfs involves minimization of the cost function $J(x)$ (*Brasseur and Jacob*, 2017):

$$J(x') = (x' - x'_a)^T S'^{-1}_a (x' - x'_a) + \gamma(y - K'x')^T S_o^{-1}(y - K'x') \tag{1}$$

where $x' = \ln(x)$ and $x'_a = \ln(x_a)$, $x_a(n \times 1)$ is the prior emission estimate (Sect. 2.2), and $y$ ($m \times 1$) is the $m$-dimensional vector of TROPOMI observations ($m = 5907939$). $S'_a$ ($n \times n$) is the prior error covariance matrix in log space and $S_o$ ($m \times m$) is the observational error covariance matrix. $K' = \partial y / \partial x'$ ($m \times n$) is the Jacobian matrix that describes the sensitivity of observations $y$ to $x'$, and $K'x' = Kx$ where $K = \partial y / \partial x$ is the sensitivity of $y$ to $x$, which can be readily represented by the GEOS-Chem forward model (*Jacob et al., 2016*).'

*It does though appear that equation (5) is derived from minimization of a cost function based on ln(x) but not the log of y. This is potentially concerning for the following reason. The authors claim that they can substitute Sa' and Ka' for Sa and K, and then directly use the equations*

*derived in Gaussian space for the posterior error (Eq 3) and averaging kernel (Eq 4). Now, I've seen and used this sort of Gaussian anamorphosis before, but only in the case wherein both the variable and observations are transformed from log-normal to Gaussian space. In this situation, indeed Eq 3 and 4 are applicable, as the solution is the BLUE solution (see equations 5.58 - 5.60 of Cohn 1997). Also see for example the application studies of Brioude 2011, Saide 2012, Cui 2019, or the theoretical works of Fletcher 2010 and Bocquet 2010, none of which, I note, are cited in the present manuscript. However, with only x transformed but not y, I'm not sure they have arrived at a BLUE solution.*

We have added a discussion on if the Gaussian transformation here could arrive at a BLUE solution on page 8 lines 304-315:

'It is of critical importance to discuss if the Gaussian transformation in Eq. (1) could arrive at a best linear unbiased estimator (BLUE) solution (*Cohn*, 1997). As $x' - x'_a \sim N(0, \mathbf{S_a}')$ and $y - \mathbf{K}'x' = y - \mathbf{K}x \sim N(0, \mathbf{S_o})$, both the prior and observational errors are Gaussian with zero mean; there is a non-linearity relationship between $x'$ and $y$ that are linked by $\mathbf{K}'$. Our analytical transformation thus conforms to the case of a 'Gaussian anamorphosis' described in *Bocquet et al* (2010), for which a BLUE solution can be properly carried out; a weak point is that the Jacobian matrix may be nonlinear, which is however sometimes the case in particular if the original Jacobian matrix is linear. Previous studies have applied this approach to transform non-Gaussian problems (*Fletcher et al.*, 2010; *Brioude et al.*, 2011; *Saide et al.*, 2015; *Cui et al.*, 2019). We acknowledge that those studies assumed non-Gaussian errors for both the prior information and the observations, while our work only assumes log-normally distributed errors on the prior state vector. '

*What is the basis for the assertion that Eq. (3) gives the posterior error of the solution to (5)? I believe it is up to the authors to show that is indeed the case, or find a paper that shows this, or if not then to demonstrate that the approximation is in some way tolerable. Citing Lu 2022 is not sufficient, as that paper makes the same assumption without discussion. Alternatively, the authors might choose instead an ensemble approach to calculating their posterior error, as we did in Cui 2019 (using ~1000 samples).*

Chapter 5.8.1.1 of *Rodgers* (2000) confirms that the solution of the non-linear problem using Levenberg-Marquardt method and Gaussian-Newton can apply the analytical posterior error calculation. We have revised and added the following text on page 8 lines 317-324:

'*Rodgers* (2000) indicated that the solution of a non-linear problem using Levenberg-Marquardt method can be applied to obtain the posterior error covariance matrix $\widehat{\mathbf{S}}'$:

$$\widehat{\mathbf{S}}' = (\gamma \mathbf{K'^T S_o^{-1} K'} + \mathbf{S'^{-1}_a})^{-1} \tag{3}$$

With the averaging kernel matrix $\mathbf{A}$ quantifies the sensitivity of the solution to the true value:

$$\mathbf{A} = \frac{\partial \hat{x}'}{\partial x'} = \mathbf{I_n} - \widehat{\mathbf{S}}' \mathbf{S'^{-1}_a} \tag{4}$$

where $\mathbf{I_n}$ is the identity matrix. The trace of $\mathbf{A}$ measures the number of independent pieces of information on $x'$ obtained from the observations and is often referred to as the degrees of freedom for signal (DOFS).'

*That being said, I appreciate the authors' ensemble / sensitivity approach to error estimation, and it could be the distinction I'm making has little impact on total error given that their error ranges from the ensemble of sensitivity runs was often larger than their analytic posterior error. But the interpretation of the error correlations and averaging kernel is rather critical to the results in the manuscript regrading sector-specific emissions, and it seems these are based entirely on the analytical calculations. Though that brings me to my last point on this topic..*

*Lastly, and separately, that the ensemble standard deviation is typically larger than the analytic standard deviation (line 331 / Fig S1) also makes me wonder about the adequacy of the analytic posterior and Ak, even if calculated correctly. Did the authors consider updating \hat{S} using information from the ensemble, rather than just the diagonal? If they in fact did do this (i.e., if Fig 6 incorporated the spread from the ensemble), it might be worth stating more clearly, as I didn't catch that.*

Thank you for the comments, and we have addressed this question on page 8 lines 323-329:

'The diagonal terms of **A** define the averaging kernel sensitivities, which quantify the extent to which the solution is informed by the observations within the inversion framework. They measure the actual error reduction if the inversion framework is correct, but errors in inversion parameters such as prior emission distributions can affect this interpretation (*Yu et al.*, 2021). An alternative and better way to estimate posterior errors is to generate an ensemble of sensitivity inversions (Sect. 2.6).'

*Minor comments:*

*97: I know what you mean, but technically it isn't "no additional" cost, though indeed it is quite minimal relative to additional CTM runs. Still, the language could be more precise.*

We have replaced 'no additional computational effort' with 'minimal added computational effort' in the revised text.

*201: The previous global inversion of Qu 2021 used for the boundary conditions was based off an earlier version of TROPOMI data. Are the biases in that version (1.02) different from the one currently used (2.01)? I know the paper discusses how biases with respect to GOSAT have changed substantially between these two, by about 20 ppb, and that is in part impacted by the regional scope and higher spatial resolution of this analysis. Still, I wonder if the use of boundary values optimized by v1.02 data that is generally biased higher than the v2.02 data contributes to in part to the emissions increases found within this region inversion. Or is this compensated for by the boundary values mostly being scaled down (Table S1)?*

We agree with the reviewer that the boundary conditions from *Qu et al.* (2021) based off an early version of TROPOMI data may be biased. We have acknowledged this point on page 5 lines 207-212 (underline part added):

'The boundary condition vertical profiles obtained from *Qu et al* (2021) avoids systematic drift of the simulation from the TROPOMI observations, but some bias could remain because *Qu et al*. (2021) used an earlier version (1.03) of the TROPOMI data and the data would not be expected to perfectly correct the model anyway. We therefore further correct the boundary

conditions on each side of our domain (north, south, west, and east) and for each season as part of the inversion (Table S1).'

*215: What is the reason for choosing 600 basis functions? This comes across as a bit arbitrary. Later we learn the inversion DOFs is 167, so 600 does seem like a safe dimension for the inversion. Still, there's no guarantee with this framework that all modes of variability in the emissions at 0.25 degree resolution captured by the observations are spanned by the 600 element GMM basis.*

We use 600 due to computational considerations and the number of DOFs in the system. And we acknowledge that we may miss some hotspots in the observations that are not present in the prior emissions. We now add the following text on page 6 lines 224-231:

'We choose to use 600 Gaussian functions, based on previous experience in inversions for North America (*Turner and Jacob*, 2015; *Maasakkers et al.*, 2021). The inversion optimizes the emission amplitude for each Gaussian. We also optimize 16 boundary conditions (four seasons × four boundaries) for a total of $n = 616$ state vector elements. Construction of the GMM does not include information from the observations and therefore might not resolve hotspots in the observations that are not present in the prior emission patterns. *Nesser et al.* (2021) proposed an alternative approach where information from the observations is integrated into the emission patterns to be optimized.'

*230: The use of the word "optimal" here is ambiguous. More precisely, the equation presented below provides the maximum likelihood estimate of the posterior (which for normally distributed errors is also the mean) and is better described as such.*

We have replaced 'optimal estimate' with 'maximum a posteriori (MAP) estimate' or 'best estimate' throughout the text.

*238: More precisely, additional regularization is used because both the observation and prior error covariance may not be well known. It's not as if gamma only corrects for unknowns in the former. In fact, your results somewhat support this. If gamma were really only compensating for misspecification of So, then we would wonder why your estimate of So is off be nearly two orders of magnitude. In contrast, if we assume some of this is owing to underestimation of Sa, that helps explain why the increments found for individual posterior x values relative to xa can exceed the prior uncertainty of 50%.*

We agree with the reviewer that $\gamma$ accounts for the missing covariance matrix in both $\mathbf{S'_a}$ and $\mathbf{S_o}$. And we have revised the text as follows (underline part added or revised):

'$\gamma$ is a regularization factor to avoid over/underfit to observations. $\gamma$ is needed because the prior and observational error covariance matrices can only be roughly estimated and is assumed here to be diagonal for lack of better information and convenience of computation.'

*241: Actually, the relationship between x and concentrations is affine, not linear, since you don't include the initial concentrations in x. But I'm guessing that the timescale of the inversion is long enough that the role of the initial conditions is minimal.*

Thank you for the comment and we have adopted this suggestion on page 6 lines 261-264 (underline part added):

'The relationship between methane emissions and concentrations in the nested GEOS-Chem simulation is linear (omitting the minimal effect of potential errors in initial conditions), so that **K** defines the forward model for the purpose of the inversion.'

We also corrected the initial conditions to minimize the mean difference with TROPOMI observations of that time (page 5 lines 202-203):

'Initial conditions on 1 January 2019 are also from the GEOS-Chem simulations by *Qu et al* (2021) and uniformly scaled to match the mean column mixing ratios retrieved from TROPOMI.'

*269: Since your emissions don't actually follow a normal distribution and thus (x-xa)^2 isn't chi-squared, does this still hold? Shouldn't it be based on ln(x) and Sa' to be chi-squared? I doubt it impacts your results though, and as you've already checked your sensitivity to gamma the overall conclusions likely remain unchanged. Still, in this case one might consider other methods for estimating gamma. By construction of So, I think your observation term is order m to begin with, so that method wouldn't work. It looks to me like the value identified in Lu 2021 Fig 4b might be at the corner of an L-curve, as the red line started to flatten again where the blue line rises dramatically.*

We wish to clarify that we did use $(x' - x'_A)^T S'^{-1}_a (x' - x'_A)$ to determine $\gamma$ but we did a poor job of communicating that message in the original manuscript. $(x' - x'_A)^2$ is chi-squared so the method we use to determine $\gamma$ is valid. We have corrected the formula on page 8 lines 330-333:

'The optimal value of $\gamma$ can be determined following *Lu et al.* (2021) so that the sum of state vector terms in the posterior estimate of the cost function, $J_a(\widehat{x'}) = (\widehat{x'} - x'_a)^T S'^{-1}_a (\widehat{x'} - x'_a)$ has value of $\sim n \pm \sqrt{2n}$, which is the expected value ($\pm 1$ standard deviation) from the Chi-square distribution with $n$ degrees of freedom.'

*291: It's not clear to me how the authors arrived at k = 10. The cited work of Lu 2022 does not explain either, and only cites Rodgers 2000. Levenberg-Marquardt schemes usually adjust k dynamically, letting it be smaller when the minimization is progressing well (i.e., approaching taking Gauss-Newton steps) and making it be larger (i.e., approaching steepest-descent steps) if the minimization starts to become unstable. How many iterations were typically required?*

We have added the following text to address this question on page 7 lines 293-303:

'$k$ is a coefficient for the iterative approach, and we tested three methods to set $\kappa$:

(1) $\kappa$ is set to 100 to start and is gradually decreased as the solution is approached, i.e., $\kappa = 101 - \max(N, 101)$ where $N$ is the iteration index;

(2) $\kappa$ is set to 100 for iterations $N \in [1,20)$; 10 for $N \in [21,40)$; 1 for $N \in [41,60)$; and 0 for $N > 60$;

(3) $\kappa$ is fixed at 10.

We find that using $\kappa = 10$ converges faster with no difference in results compared to the other two methods, and adopt that method in what follows. We iterate on Eq. (2) until the maximum difference in state vector elements between two consecutive iterations ($x'_N$ and $x'_{N+1}$) is smaller than 0.5%, at which point we adopt $\hat{x}' = x'_{N+1}$ as the best posterior estimate. It takes the base inversion 139 iterations to converge to the solution.'

*281 - 305: I note that discussion of Equation 5 (lines 281 - 289, then again 295-305) conceptually follows Lu 2022 sentence by sentence, which while paraphrased, is not original content. Would it be better to just cite Lu 2022 here rather than repeat? Otherwise we have about a dozen instances in sequence like the following:*

*Lu 2022: The boundary conditions are still optimized with normal error distributions, assuming an error standard deviation of 10 ppb.*

*Chen 2022: The normal error assumption is retained for the boundary conditions elements of the state vector, with a prior error standard deviation of 10 ppb.*

*Perhaps this is a moot point though, if the authors do revise this section given some of the questions raised earlier.*

Thank you for the suggestion. We have re-constructed Sect. 2.4 in the revised manuscript, in which we shortened or deleted these sentences.

*Throughout: could use some more proofreading, for example*

*14: as a prior*

*23: the uncertainty range*

*34: unaccounted for (?)*

*84: extra space*

*85: missing spaces*

Corrected as suggested.

**RC2: Anonymous Referee #1**

*The manuscript by Chen et al. presents the estimate of methane emissions for 2019 in China by using the TROPOMI data, an inversion method, and GEOS-Chem model simulations. Overall, paper is interesting and could be published after the following concerns are addressed.*

We thank the reviewer for the constructive comments. All points have been addressed as below.

*1.Uncertainty in the abstract and the text. Should the uncertainty range be centered at the best estimate? In some cases, it is; CH4 from coal is 16.6 with uncertainty range of 15.6-17.6. In some case, it is not; total anthropogenic emission is 65 with uncertainty range of 57.7 - 68.4. Please explain why it is so, and how the uncertainty is defined? Is it because the errors from the prior emission are assumed to be log-normal? but this still doesn't explain why in some cases, the best estimate is indeed centered at the uncertainty range.*

The uncertainty range are generally from inversion ensemble and is not necessarily centered at the best estimate. We have clarified the uncertainty range in the abstract (underline part added):

'Our best estimate for total anthropogenic emissions in China is 65.0 (57.7-68.4) Tg a$^{-1}$, where parentheses indicate the uncertainty range determined by the ensemble.'

We also specifically described the uncertainty range in the last paragraph of Sect. 2.6:

'The uncertainty in posterior estimates reported here is taken as the greater of the range of solutions given by the inversion ensemble and the 2-sigma error inferred from the diagonal of $\hat{\mathbf{S}}$, and is generally determined by the ensemble (Fig. S1).'

*2.The motivation of this paper should be strengthened in the introduction section. It appears that Qu et al (2021) already done global estimate of CH4 emission. How does the results from that paper compare with bottom-up estimate in China and motivate this work? Is it because Qu et al didn't do sector attribution at fine spatial resolution? More articulation above L90 is needed here.*

Thank you for the suggestion and we have strengthened our motivation on page 3 lines 89-92:

'*Qu et al.* (2021) pointed out that their TROPOMI inversion suffered from major artifacts in southern China due to mislocation of prior coal emissions, juxtaposition of coal and rice emissions at the ~200 km resolution of the inversion, and extensive seasonal cloudiness.'

*3.Bias in boundary conditions (text around L200). While the importance of the boundary conditions is recognized here, does the optimization here attribute the innovation to the emissions within China only or also emissions in elsewhere as well? How are the boundaries defined and will the results be sensitive to the location of boundaries? Methane is long lived, and model-observation difference in CH4 can be attributed to the upper wind regions that are far from the domain of interest, even at the seasonal scale.*

Thank you for the comment. We use boundary conditions from the global GEOS-Chem vertical profiles optimized by *Qu et al* (2021) and the vertical profiles would factor in the contribution

from the upper wind regions. We have clarified the boundary conditions on page xx line xx (underline part revised):

'The nested version of GEOS-Chem is similar to that used in previous regional inversions of TROPOMI observations (Y. *Zhang et al.*, 2020; *Shen et al.*, 2021) and uses 3-hour dynamic boundary conditions from the global GEOS-Chem simulated vertical profiles at $2^\circ \times 2.5^\circ$ resolution for 2019 with posterior methane emissions optimized by TROPOMI observations (*Qu et al.*, 2021).'

We have added the definition of boundary conditions on page 3 lines 122-125 (underline part added):

'Global mean bias in the TROPOMI observations is inconsequential for regional inversions because it can be incorporated in the boundary conditions (defined as the edges of the study domain), and random error (precision) is effectively reduced through the large number of observations (Fig. 1).'

*4.Section 2.4. Does the state vector include sector resolved emissions or simply spatially resolved total emissions? 'total' could be added here. Why number of Gaussian functions is 600 ? Later, 'the inversion optimizes the emissions for each Gaussian function along with 16 boundary conditions ... for a total of .. 616 elements". Should the inversion optimize the state vector x? Consider to add 'in the state vector' at the end of this sentence. Also, should all three Gaussian parameters be optimized, and so there would be 3\*600 +16 = 1816 elements?*

We clarify in the revised manuscript that (1) parameters to build Gaussians are determined before the analytical inversion is conducted, and (2) the state vector elements in the inversion are scaling factors adjusting the amplitudes of 600-member Gaussians. We have added the following text to Sect. 2.4:

'Specifically, we project methane emissions at $0.25^\circ \times 0.3125^\circ$ resolution onto *K*-dimensional Gaussian functions where *K* is the number of similarity criteria on the $0.25^\circ \times 0.3125^\circ$ grid, in this case 14 similarity factors including longitude and latitude (spatial proximity), and the prior emission patterns by sector (Sect. 2.2). Each multivariate Gaussian is hence built to characterize the location (determined by longitude and latitude), emission magnitude and distribution from different sectors (*Turner and Jacob*, 2015). The parameters of the Gaussians are estimated using an expectation-maximization algorithm (*Dempster et al.*, 1977) to find the maximum likelihood. We choose to use 600 Gaussian functions, based on previous experience in inversions for North America (*Turner and Jacob*, 2015; *Maasakkers et al.*, 2021). The inversion optimizes the amplitude for each Gaussian. We also optimize 16 boundary conditions (four seasons × four boundaries) for a total of *n* = 616 state vector elements. Construction of the GMM does not include information from the observations and therefore might not resolve hotspots in the observations that are not present in the prior emission patterns. *Nesser et al.* (2021) proposed an alternative approach where information from the observations is integrated into the emission patterns to be optimized.'

*Otherwise, will it be possible that the new source of CH4 in the regions where the prior emissions are low can not be accurately located in the domain with GMM? The fine-resolution TROPOMI data has the potential to accurately identify the local new sources. GMM appears to have the limitation to locate a new source spatially in areas that the prior emission informs the possibility for large-spatial aggregation. True?*

We agree with the reviewer on the potential limitation from merging weak emissions. The same issue is also raised by the other reviewer. We have acknowledged this limitation and suggested a future solution on page 6 lines 227-231:

'Construction of the GMM does not include information from the observations and therefore might not resolve hotspots in the observations that are not present in the prior emission patterns. *Nesser et al*. (2021) proposed an alternative approach where information from the observations is integrated into the emission patterns to be optimized.'

*5.Aggregation vs. smoothness. From Figure 4c, it appears there are still distinct boundaries in post/prior ratios, which could suggest an artifact as a result of aggregation. I would recommend to plot it with a different color bar that has more smooth transition of the colors and discuss further if there are still distinct boundaries.*

Thank you for the suggestion and we have revised Fig. 4c as suggested:

[Figure]

**Figure 4**. Optimization of methane emissions over East Asia in 2019 from inversion of TROPOMI observations. Results are from the base inversion and are shown on the 0.25°× 0.3125° grid. (a) Prior estimates of methane emissions, summing the contributions from the sectors in Fig. 3 plus additional minor sectors as given in Table 1. (b) Posterior methane emissions from the TROPOMI inversion. (c) Posterior/prior emission ratios. (d) Averaging kernel sensitivities. The averaging kernel sensitivities are the diagonal elements of the averaging kernel matrix and display the ability of the observations to quantify emissions independently from the prior estimates (1 = fully, 0 = not at all). The degrees of freedom for signal (DOFS, defined as the trace of the averaging kernel matrix) is given inset.

*6.Gaussian representation. 'a mean location, spatial standard deviation, and emission magnitude'. How are those parameters defined by using the a prori emission? What does the mean location mean? Is it emission-weighted geographic center?*

This has now been clarified in our response to comment #4 above. The mean location is estimated using longitude and latitude as part of the GMM to build Gaussians and is not the emission-weighted geographic center.

*7.L240. 'the relationship between methane emissions and concentrations in the nested GEOS-Chem simulation is linear'. Is this an assumption? In what time scale? I would think even there is no methane emission in China, global transport would still result in the methane in the air in China.*

Thanks for catching this and we have corrected this point on page 6 lines 261-265 (underline part added).

The relationship between methane emissions and concentrations (or more precisely, concentration enhancement above background mixing ratios) in the nested GEOS-Chem simulation is linear (omitting the minimal effect of potential errors in initial conditions), so that **K** defines the forward model for the purpose of the inversion.

*8.Section 2.5. The sectoral attribution. How is the summation matrix W defined? In addition, from what can tell here, it seems that the attribution yields the single scaling factor for the national emission from each sector. In other words, it didn't spatially resolve the scaling factor for each sector in every aggregated area. will it make sense that the sectoral attribution is made in proportion of the relative weights of each sector from the prior emission in every aggregated area? Description of the assumption or reasons for the feasibility of the attribution method is needed here.*

Our inversion can spatially resolve the scaling factor for each sector in the aggregated area as displayed in Fig. 4c. We have extended the description of the summation matrix construction and elaborated on this point in Sect. 2.5:

'Here we use the reduction of the state vector to 12 sectors (Sect. 2.2) of aggregated emissions as an example to illustrate the construction of **W**. The GMM approach derives the relative weighting of each Gaussian on the $p$ native-resolution grid $\mathbf{W_1}$ ($p \times n$); $\mathbf{W_1}$ thus allows the spatial allocation of posterior state vector to individual grids (Figs. 4c-d). $\mathbf{W_1}$ is further multiplied by $\mathbf{W_2}$ ($12 \times p$), the fractional contribution of individual grid by sector to total emissions, to obtain the summation matrix $\mathbf{W} = \mathbf{W_2}\mathbf{W_1}$ ($12 \times n$ in this particular example).'

*9.Section 4.1. I could miss something here. but, is there any map showing the change of emission by sector? so it can support that 'our downward correction of .... is driven by the Shanxi province and southwestern China'? This is also relevant to my comments in #4 and #8 as well as the abstract - 'our higher livestock emissions are attributed ... to northern China'. A map showing the posterior/prior ratio for each sector can be useful in the main text.*

Figure 4c shows the posterior/prior ratios mapped onto the native-resolution grid for any sector. The response to comment #8 has partially addressed this point, and we have further added clarification on page 11 lines 425-427 (underline part added):

'Sectoral attribution assumes that the posterior/prior emission ratios for a given 0.25°×0.3125° grid cell (Fig. 4c) apply equally to all prior emission sectors within that grid cell, so that the combination of Fig. 4c and Fig. 3 gives the spatial distribution of the change in emission by sector.'

*Minor.*

*In abstract. mention the year for the prior emission.*

Added as suggested.

**References (newly added):**

Bocquet, M., Pires, C. A., & Wu, L.: Beyond Gaussian statistical modeling in geophysical data assimilation. Monthly Weather Review, 138(8), 2997–3023. https://doi.org/10.1175/2010MWR3164.1, 2010.

Brioude, J., Kim, S.W., Angevine, W.M., Frost, G.J., Lee, S.H., McKeen, S.A., Trainer, M., Fehsenfeld, F.C., Holloway, J.S., Ryerson, T.B. and Williams, E.J: Top-down estimate of anthropogenic emission inventories and their interannual variability in Houston using a mesoscale inverse modeling technique, J. Geophys. Res., 116, D20305, doi:10.1029/2011JD01621, 2011.

Cohn, S. E.: An introduction to estimation theory. J. Meteor. Soc. Japan, 75, 257–288, https://doi.org/10.2151/jmsj1965.75.1B_257, 1997.

Cui, Y., D. K. Henze, J. Brioude, W. M. Angevine, Z. Liu, N. Bousserez, J. Guerrette, S. A. McKeen, J. Peischl, B. Yuan, T. Ryerson, G. Frost, M. Trainer: Inversion estimates of lognormally distributed methane emission fluxes from the Haynesville–Bossier oil and gas production region using airborne measurements, J. Geophys. Res., 124, 3520–3531, https://doi.org/10.1029/2018JD029489, 2019.

Dempster, A. P., Laird, N. M., and Rubin, D. B.: Maximum likelihood from incomplete data via the EM Algorithm, J. Roy. Stat. Soc. B, 39, 1–38, https://doi.org/10.1111/j.2517-6161.1977.tb01600.x, 1977.

Fletcher, S. J.: Mixed Gaussian lognormal four dimensional data assimilation. Tellus, 62 (3), 266–287, https://doi.org/10.1111/j.1600-0870.2009.00439.x, 2010.

Nesser, H., Jacob, D. J., Maasakkers, J. D., Scarpelli, T. R., Sulprizio, M. P., Zhang, Y., and Rycroft, C. H.: Reduced-cost construction of Jacobian matrices for high-resolution inversions of satellite observations of atmospheric composition, Atmos. Meas. Tech., 14, 5521–5534, https://doi.org/10.5194/amt-14-5521-2021, 2021.

Saide, P., Peterson, D. A., da Silva, A., Anderson, B., Ziemba, L. D., Diskin, G., Sachse, G., Hair, J., Butler, C., Fenn, M., Jimenez, J. L., Campuzano-Jost, P., Perring, A. E., Schwarz, J. P., Markovic, M. Z., Russell, P., Redemann, J., Shinozuka, Y., Streets, D. G., Yan, F., Dibb, J., Yokelson, R., Toon, O. B., Hyer, E., & Carmichael, G. R.: Revealing important nocturnal and day-to-day variations in fire smoke emissions through a multiplatform inversion. Geophysical Research Letters, 42, 3609–3618. https://doi.org/10.1002/2015GL063737, 2015.

Yu, X., Millet, D. B., and Henze, D. K.: How well can inverse analyses of high-resolution satellite data resolve heterogeneous methane fluxes? Observing system simulation experiments with the GEOS-Chem adjoint model (v35), Geosci. Model Dev., 14, 7775–7793, https://doi.org/10.5194/gmd-14-7775-2021, 2021.

---

## Author Response (AR2)

We thank the reviewer for the comments on the manuscript. Below, we have replied to the review and have detailed the corresponding edits that we have made to the manuscript. We have listed out the reviewer comments in *black italic* and the replies in blue.

**RC1: Dr. Daven Henze**

*Overall, the manuscript revisions address nearly all of the questions raised by myself and other reviewers, which were mostly issues of clarification, and I appreciate their efforts and explanations.*

*I do have one outstanding point / question: — the authors assert that y - K'x' = y - Kx ~N(0,So), i.e. the measurements follow a normal distribution with covariance So. Can they provide a plot to show this? Normality here is helpful as the framework after the x to x' transformation appears Gaussian and traditional equations apply for linear estimates of the posterior error at the solution, and the Ak. It is still a bit contradictory though to say that x is log normally distributed, K linearly well represents the forward model, and yet that y - Kx is normally distributed. It's possibly very close though. So, certainly it is easy to check — can they make a histogram of y - Kx, or even just y, and show that it is normally distributed? I know with some remote sensing datasets there will be a subset of y that is actually negative, consistent with a normal distribution. This can pose a bit of a quandary though if one filters the negative data, because then the distribution is no longer Gaussian. So I wonder if the authors encounter this at all, and, if they do, how it was dealt with.*

*Regardless, this is a rather small point of discussion, given that the authors recognize the limitations of the analytical expressions themselves and incorporate ensemble information into their error estimates. I'm only belaboring this point as it applies to a methodology this group is using for multiple studies. I trust they can check / report the skewness of their data and make updates to the manuscript as appropriate without further need for review.*

We thank the reviewer for the suggestion. We have added a figure (Fig. S1) in the Supporting Information to show that $y - K\hat{x}$ is normally distributed, and denoted it on Page 7 lines 298-300 (underline part added):

'As $x' - x'_a \sim N(0, S'_a)$ and $y - K'x' = y - Kx \sim N(0, S_o)$ (Fig. S1), both the prior and observational errors are Gaussian with zero mean; there is a non-linearity relationship between $x'$ and $y$ that are linked by $K'$.'

[Figure]

**Figure S1**. The histogram distribution of $\boldsymbol{y} - \mathbf{K}\hat{\boldsymbol{x}}$.